# Adaptive Real-Time Channel Estimation and Parameter Adjustment for LoRa Networks in Dynamic IoT Environments

**DOI:** 10.3390/s25072121

**Published:** 2025-03-27

**Authors:** Fatimah Alghamdi, Fuad Bajaber

**Affiliations:** Department of Information Technology, Faculty of Computing and Information Technology, King Abdulaziz University, Jeddah 21589, Saudi Arabia; fbajaber@kau.edu.sa

**Keywords:** channel state estimation, LoRa networks, multi-task LSTM, real-time processing, adaptive parameter adjustment, IoT, dynamic environments, overlap window, online incremental learning, hybrid feature extraction, confidence-based adaptation, wireless communication

## Abstract

This study addresses the challenges of real-time channel state estimation and adaptive parameter adjustment in dynamic LoRa networks, where the existing methods often fail to adapt efficiently to highly variable channel conditions. This study presents an innovative approach for real-time channel state estimation and adaptive parameter adjustment in long-range (LoRa) networks in dynamic Internet of Things (IoT) environments. When these types of networks are used in dynamic IoT environments, they are known to face challenges in the two above-mentioned areas. In our approach, a hybrid feature extraction method that integrates statistical analysis with domain-specific knowledge is utilized for real-time data labeling, focusing on the signal-to-noise (SNR) and received signal strength indicator (RSSI) metrics. This approach employs an adaptive sliding window technique for efficient processing of recent data. Subsequently, a multi-task long short-term memory (LSTM) neural network is introduced for the simultaneous prediction of multiple channel states. This multi-task model employs an online incremental learning approach to enhance the real-time performance and responsiveness of the model within dynamic environments. It also incorporates a confidence measure for estimated states to increase the prediction reliability. Finally, based on the confidence measure predictions and channel state estimation, the system dynamically adjusts the LoRa parameters, including the spreading factor, coding rate, transmission power, and bandwidth. Our results demonstrate that the confidence-based adaptive strategy coupled with adaptive sliding window processing and incremental learning effectively balances performance optimization with stability in challenging IoT scenarios. This study contributes a robust, data-driven approach for real-time channel state estimation and adaptive parameter control, addressing the unique challenges of IoT networks in dynamic environments. Our approach achieved a packet delivery ratio of 100%, reduced energy consumption to 0.07987 Joules per packet, and demonstrated a prediction accuracy between 97.70% and 97.9% for estimating the different channel states. This innovative framework provides significant improvements in channel state estimation, communication reliability, adaptive parameter control, and computational efficiency, thereby ensuring robust performance in IoT environments at the same time.

## 1. Introduction

The rapid proliferation of Internet of Things (IoT) devices has led to increased demand for reliable and efficient wireless communication technologies. Due to its extended reach and relatively low power requirements, the long-range (LoRa) communication technology has become a strong contender for various IoT applications. Nevertheless, various dynamic channel conditions, such as fading, shadowing, interference, path loss, and noise, can significantly degrade the performance of LoRa networks. However, methods to improve network performance, such as classic channel state estimation and parameter adjustment, are ineffective as they cannot adjust rapidly to different conditions, resulting in degraded performance and limited reliability. Moreover, most wireless sensor network (WSN) impairments have similar characteristics and are captured through the monitoring of SNR and RSSI values, with less attention being paid to the underlying mechanisms. Notably, each impairment has its own specific LoRa parameters that can be adjusted to alter its impact [1,2].

Problem statement: The most critical aspect of LoRa-based IoT deployments is the rapid adjustment of channel parameters based on the channel state; thus, maintaining reliable communications can be very challenging. Conventional channel state estimation typically allows for the forecasting of a variety of channel phenomena but not necessarily accurately or quickly enough for real-time use in this context. In addition, existing parameter tuning methods are often based on rules or slow-adapting heuristic algorithms that do not adequately adapt to changes in the environment. An adaptive system that can estimate the different channel states in real time and intelligently to determine the best parameter variations is essential for providing optimal performance and reliability [1,2]. Existing channel state estimation and prediction methods for dynamic wireless networks, particularly in the context of LoRa systems, face several challenges:Lack of a unified multi-tasking approach.There is a critical need for a unified multi-tasking approach that can efficiently address multiple challenges in IoT and wireless sensor network (WSN) environments simultaneously while minimizing resource utilization [3].Accuracy in dynamic environments. Existing approaches struggle to accurately predict channel states under rapidly changing conditions, especially in WSNs, where sensor movements and environmental factors can cause frequent fluctuations [4,5].Real-time processing. Many existing solutions are computationally intensive, making real-time processing and adaptation difficult, especially for resource-constrained devices [1,6].Comprehensive feature extraction. There is a need for more effective methods to extract both temporal and frequency-domain features from channel state information in order to capture the main phenomena leading to network degradation [7,8].Adaptive parameter optimization. Existing methods often lack the ability to dynamically adjust the communication parameters based on real-time channel conditions, leading to sub-optimal performance [1,2]. The adaptive data rate (ADR) mechanism in a LoRa network aims to address this problem; however, its performance can be severely affected in the context of highly dynamic wireless channels [9,10].Energy efficiency. The computational complexity of current deep learning approaches can be considered a limitation for energy-constrained WBAN devices [1,8,11].Interference management.LoRa systems are particularly susceptible to channel fading and interference, which significantly impact their signal detection performance [5,12].Capture of long-term temporal dependencies. The dynamic nature of LoRa networks in environments involving WBANs or IoT devices presents significant challenges in terms of capturing long-term temporal dependencies while adapting quickly to changing conditions [5].Balance between network adaptability and stability in dynamic environments. Rapid parameter adjustments can lead to network oscillation and performance degradation [9,13], necessitating a confidence-based approach which ensures that parameters are modified only when predictions meet a sufficient reliability threshold, thereby preventing unnecessary fluctuations while maintaining the system’s responsiveness to genuine environmental changes [11].Balancing computational efficiency and accuracy. While deep learning approaches have shown promise in improving channel estimation and signal detection, they often come with high computational complexity. This poses a challenge for their implementation in energy-constrained IoT devices [10].

To address these challenges, there is a need for a comprehensive framework that combines advanced channel state estimation techniques with reliable and adaptive parameter optimization. Such a framework should be capable of accurate prediction in dynamic environments, real-time processing, and effective interference management while maintaining energy efficiency for WSN applications. In our approach, these challenges, including the need for unified multi-tasking, real-time processing, dynamic adaptability, and energy-efficient solutions, are collectively addressed in the proposed multi-task LSTM, which offers a robust and comprehensive framework to enhance the performance and reliability of LoRa networks in dynamic IoT environments.

Research motivation: With rapid progress in various technological areas, such as the IoT and smart cities, the demand for additional reliability and efficiency in wireless communications is increasing. In particular, there is more interest in time-varying environments, such as WSNs, and the use of LoRa systems, which require advances in channel state estimation and prediction features. However, existing methods cannot accurately predict the channel states in fast time-varying channels, or they have difficulties associated with issues such as interference management and energy efficiency. Computational complexity and failure to adapt to the real lead times of actual wireless systems can render current approaches ineffective for use in critical applications. This inspired us to design a holistic approach that can exploit the ability of deep learning—particularly long short-term memory (LSTM) networks—to model temporal dependencies and long-range temporal correlations in channel dynamics in order to address these challenges. Continuous online learning allows the model to constantly update new data, facilitating its evolution and ability to quickly capture relevant dependencies and correlations in a real-time environment. For this reason, using a sliding window technique is helpful as it avoids excess overhead by breaking down the current dataset into more manageable portions and avoids unnecessary recompilation of features when the model is within an existing time period that provides no new, potentially useful data available for computation. It incorporates prediction confidence into the fine-tuning of the LoRa parameters through gradual adjustment, which is important for stabilizing network performance by avoiding continuous fluctuations when there is insufficient knowledge of confidence in the predictions within a degree of commitment. This promotes efficient communication that strikes a balance between responsiveness and stability under dynamic circumstances. Responsiveness is prioritized when adjustments are critical for optimizing a network’s performance under changing conditions. Stability is emphasized when adjustments are unnecessary to avoid over-adjustment and maintain steady performance. This balance ensures efficient communication and minimizes unnecessary fluctuations in dynamic environments, which are key to maintaining reliable LoRa network operations and resource conservation.

Research scope; In this study, we propose a new multi-task LSTM-based framework for the estimation and prediction of channel states in dynamic wireless environments that specialize in LoRa systems and WSNs, including the design of a novel hybrid feature extraction approach that effectively extracts features containing empirical data and knowledge of the impairment characteristics to comprehensively represent complex channel dynamics. We focus on developing specific techniques for real-time processing through adopting an adaptive sliding window with online incremental learning to keep pace with the dynamic nature of channel conditions. Moreover, this research examines a combination of confidence-based adaptation mechanisms and the design of a predictive intelligent system that can dynamically regulate the LoRa parameters depending on the channel states. This study presents a novel framework and provides an extensive evaluation using real wireless network traces to demonstrate that the proposed approach outperforms traditional approaches in terms of prediction accuracy, communication reliability, and computational burden.

Research importance: This study addresses important problems in the rapidly developing areas of IoT and wireless communications that are relevant to LoRa networks. This work is particularly crucial as it can significantly improve the robustness and efficiency of IoT device deployment in changing landscapes. The contributions of this study are threefold. First, a hybrid feature extraction method is used along with an adaptive sliding window to capture short- and long-term dependencies, which provides a holistic view of channel behaviors. Using a sliding window with a step size equal to 1 helps to speed up the detection of short-term fluctuations, which makes it suitable for real-time monitoring. Second, a multi-task LSTM model combined with an incremental online learning approach is considered as a new method to cope with the complexity of wireless channel dynamics. As IoT applications expand to new, more complex, and sometimes extreme environments, maintaining robust network connectivity is a key factor. In addition, especially when considering adaptive systems, the confidence measures for channel state predictions provide an additional layer of reliability that allows for the minimization of errors in parameter tuning, which can otherwise cause the system to experience communication failures. Third, LoRa parameter adjustment based on the predicated channel state and confidence measure enables gradual adaption based on channel state complexity, which leads to the migration of unnecessary adjustments that maintain the balance between adaptability, reliability, and consistency of a network’s performance over time. This study provides a basis for more robust adaptive wireless communication systems in the future. This is expected to bring benefits to the growing areas of intelligent transportation systems and smart cities, as well as in a variety of other related domains, such as industrial IoT, environmental monitoring, and healthcare applications.

### 1.1. Objectives

In this study, we propose a multi-task LSTM model which employs a developed real-world labeling dataset for real-time channel state estimation and corresponding adaptive parameter control under dynamic IoT scenarios applied to LoRa communication systems. The research objectives are as follows:To create a reliable and diverse set of labeled real-time channel state data for LoRa networks. The aim was to obtain a dataset which captures the complex dynamics of wireless channels, including fading, shadowing, interference, noise, path loss, and attenuation.To develop a real-time multi-task channel state estimation model which is capable of simultaneously predicting fading, shadowing, interference, noise, path loss, and attenuation in LoRa networks.To enhance the model’s ability to process recent data efficiently and adapt continuously to changing channel conditions through the implementation of a sliding window approach with incremental online learning.To capture both long- and short-term dependencies of channel behaviors through the implementation of an adaptive sliding window with a step size equal to 1.To improve the reliability and effectiveness of the parameter adjustments by incorporating a prediction confidence measure for each predicted channel state.To optimize LoRa communication parameters in real time through the design of an adaptive parameter adjustment system that utilizes high-confidence channel state predictions.To evaluate the performance of the proposed system in terms of resource conservation, communication reliability, and computational efficiency compared to existing methods.

### 1.2. Contributions

This research makes several important contributions to the literature in the areas of the IoT and wireless communications, as follows:The statistical analysis is complemented with domain-specific knowledge to create a more complex representation of the channel dynamics than fully data-driven or rule-based methods.A multi-task LSTM neural network that considers multiple channel state values jointly is proposed to obtain a wider and better perspective of the wireless environment and channel state.To overcome the shortcomings of existing methods, we utilize the adaptive sliding window approach with online incremental learning to support real-time continuous learning from dynamical channel conditions when the method is in operation.We propose a new adjustment mechanism that utilizes a confidence measure for each predicted channel state, allowing for more reliable and guided parameter adaptation. This methodology improves the system’s adaptability in maintaining the convergence performance and stability in highly dynamic environments.We propose an intelligent approach to dynamically adjust different parameters using knowledge from high-confidence predictions of the channel states and achieve robust performance against various events.We performed an extensive process analysis using real-world data of the proposed bountiful communication scheme. The proposed model shows remarkable accuracy in terms of prediction improvement, high reliability in wireless network prediction, and very competitive computational performance compared to other state-of-the-art techniques.We create a comprehensive real-time dataset that combines statistical analysis with domain-specific knowledge, allowing for accurate labeling of various channel states.

This set of contributions advances the state of the art in adaptive wireless communication for IoT applications as a comprehensive solution for reliable transmission under time-varying conditions.

This research article is systematically organized into several key sections, providing a comprehensive understanding of this study. Section 2 delves into the foundational aspects, covering topics such as the challenges faced by existing technologies, adaptive sliding window approaches, multi-task long short-term memory (LSTM) models, prediction confidence, adaptive characteristics, and the adjustment of LoRa parameters. Following Section 2, Section 3 provides a thorough review of the existing literature, setting the stage for the novel contributions of this research. In Section 4, a multi-task LSTM neural network is proposed, which considers multiple channel state values jointly to gain a broader perspective on the wireless environment. An adaptive sliding window approach, combined with online incremental learning, supports real-time continuous learning under dynamic channel conditions. The statistical analysis is complemented with domain-specific knowledge to create a more complex representation of channel dynamics. A new adjustment mechanism utilizes confidence measures for each predicted channel state, allowing for more reliable and guided parameter adaptation. This intelligent approach dynamically adjusts different parameters, achieving robust performance against various events. Section 5 details an extensive process analysis using real-world data. The proposed communication scheme shows remarkable accuracy in terms of prediction improvement, high reliability in wireless network prediction, and competitive computational performance when compared to other state-of-the-art techniques. Finally, Section 6 summarizes the contributions of this study, highlighting its impacts for advancing adaptive wireless communications for IoT applications and its potential benefits in various related domains, such as intelligent transportation systems, smart cities, industrial IoT, environmental monitoring, and healthcare applications.

## 2. Background

In this section, we outline the important terminology and techniques used for channel state estimation and LoRa adjustment. We demonstrate the adaptive capabilities of LoRa communication, the adaptive sliding window technique, incremental learning, multi-task LSTM, and confidence prediction.

### 2.1. Challenges of Existing LoRa Datasets

Most currently available LoRa datasets have significant drawbacks, which means that they might not be effective for tasks such as estimating different states of LoRa channels at the same time when using a multi-task LSTM. These challenges include the following:Absence of holistic environmental data. Even with datasets that contain topographic data and bottom boundary roughness, detailed environmental data are often absent, limiting the ability to use these data to understand how the state of a channel may vary. In [14], it was noted that “most existing datasets do not contain fine-grained environmental data such as node proximity temperature, which is known to significantly impact communication performance”. Furthermore, the LoRa-at-the-edge dataset (LOED) [15] offers valuable data for analyzing LoRa networks but does not provide detailed environmental information, such as distance and spatial complexity. These datasets often miss out on more detailed data that could greatly improve the understanding and prediction of channel state variations. This restriction prevents the dataset from being used to extract complex patterns in the context of channel state estimation.Low temporal resolution. While short-term variations in the channel state are important, most current datasets lack high-frequency samplings. As an example, the dataset of Cardell-Oliver et al. [16] consists of “half hourly readings”; thus, its spatial granularity may be too coarse to detect rapid fluctuations in the channel state.Limited network scale. In existing datasets, the information was typically retrieved from only a few nodes, which may not be sufficiently complex to characterize larger LoRa networks. For example, the dataset of Cardell-Oliver et al. [16] only includes data from “5 transmitters and two receivers”, which may not address the variety of channel conditions found in larger environments.Insufficient long-term data. Some datasets do not include long-term data collection, which is necessary to identify seasonal and long-term trends in a channel. Many such collections only gather data over a few days of operation, whereas some datasets—such as that of Tian et al. [14]—contain information gathered for months.Apparent limitations in physical layer information. The datasets obtained by Tian et al. [14] focused on “network-level performance (e.g., the average number of correctly exchanged packets)”, which may not provide sufficient information to achieve precise channel state estimation.Most existing datasets [14,15,16] provide valuable radio frequency parameters such as RSSI and SNR, as well as LoRa configuration parameters such as the spreading factor (SF), bandwidth (BW), coding rate (CR), and frequency, which are vital for a comprehensive understanding of the network’s analytical context; however, a significant issue is the lack of labeled data, which makes these data difficult to use for supervised learning approaches to channel state estimation.

To overcome these obstacles, we performed a hybrid feature extraction process that involved statistical analysis and domain-specific knowledge to capture binary classification features and continuous value features in real time. The proposed hybrid feature extraction method analyzes the SNR and RSSI values to capture different parameters in order to estimate different channel states.

### 2.2. Adaptive Sliding Window and Incremental Learning Technique

The sliding window technique is a common approach for addressing the time-varying nature of data in dynamic systems. Sliding windows are used in dynamic networks to keep track of the variations in channel conditions and hold this information for IoT and WSN applications. In particular, this is important for LoRa communications in time-varying environments [17]. Sliding windows facilitate adaptive channel estimation by emphasizing recent channel information while ignoring older (and possibly stale) observations. This method is appropriate considering the ever-changing nature of IoT and WSN deployments, as channel conditions can significantly alter mobility and environmental changes [17].

An adaptive sliding window approach [18] is another type of sliding window technique that allows for analysis of the variability of the signal strength at different time intervals and with different window sizes based on predefined events. This allows both long- and short-term variations to be captured, reduces the computational overhead, and reduces latency.

Incremental learning is a machine learning paradigm in which models can gradually adapt to new data and learn better as more feedback emerges. This is a very important method not only in IoT and WSN applications but also in LoRa networks, as the communication environment evolves continuously [19]. The incremental learning model for the channel estimation task can deal with changing channel conditions in a real-time manner. The accuracy of the estimate of the new channel measurements can be gradually improved without complete retraining [19].

The utilized adaptive sliding window and incremental learning approaches have various advantages for IoT and WSN applications, which are now more apparent in the context of LoRa communications:A sliding window ensures lower overhead as the amount of data to be processed is reduced when compared to processing the entire history, reducing the burden associated with channel estimation [17].Lower latency: Incremental learning allows for rapid updating of channel estimators, enabling real-time adaptation to changing conditions without introducing significant processing delays [20].More accurate estimation: The adaptive nature of updating channel estimates with the latest measurements improves the estimation accuracy and reliability over time, hence providing accurate and timely channel state information (CSI) [17,21].Adaptation to changing environments: IoT and WSN deployments typically feature mobile nodes and dynamic surroundings. The changes can be well monitored through the use of sliding windows and incremental learning [17].An adaptive sliding window with a step size of 1 can effectively capture short-term variations immediately while simultaneously gathering information reflecting long-term variations.

### 2.3. Multi-Task Long Short-Term Memory (LSTM)

In recent years, long short-term memory (LSTM)-based neural networks have become increasingly popular for the estimation of channel state information (CSI) in wireless communication systems. LSTM networks are considered more appropriate for time-series data as they can capture long-term dependencies. This ability is essential in the context of wireless communications, where channel conditions can fluctuate rapidly over time. CSI estimation and related multi-task learning (MTL) approaches based on LSTM networks have shown impressive potential. In [3], the authors utilized an MTL framework for joint CSI prediction and a predictive transmitter selection strategy for use in mobile communication scenarios. The approach in [3] is based on a single LSTM network architecture that jointly learns two tasks: CSI prediction and transmitter selection. This multi-task method showed significantly better accuracy than sequential task-learning models, especially in cases where there are large variations in the number of transmitters and mobile node speeds [3]. There are many benefits associated with using multi-task LSTM models for CSI estimation:Multi-task LSTM models exhibit better performance compared to traditional approaches, such as least squares and minimum mean square error channel state estimation. This includes higher accuracy, especially in cases with unknown channel statistics [3].LSTM networks can capture temporal dependencies in data. This property is extremely important for CSI estimation as wireless channels are time-varying [3]. Analyzing CSI from the past and predicting what it would look like in the future allows for communication systems that are stronger and more resilient to change.Multi-task LSTM approaches can offer significant computational and memory efficiency. The authors of [3] reported a substantial reduction in computational time (approximately 40%) associated with their MTL architecture. Such efficiency is particularly valuable for resource-constrained wireless systems [3].LSTM-based estimators have been shown to be much more adaptive than classical model-based estimation methods. This flexibility enables them to achieve good performance under different channel conditions, including noisy and interference wireless channels.Deep learning approaches, including LSTM-based multi-task models, can handle large amounts of data, detect statistical dependencies and features in the data, and generalize to new datasets that were previously unseen. Due to this generalization ability, multi-task LSTM-based estimators are potential candidates for 5G-and-beyond communication systems, which need to adapt to various channel conditions that may continuously change over time.

### 2.4. Prediction Confidence

Previous studies have explored various methods for channel estimation and prediction in wireless communication systems, particularly orthogonal frequency-division multiplexing (OFDM). Sternad and Aronsson [22] proposed the use of Kalman filtering and prediction for accurate channel estimation in adaptive OFDM systems. This approach formed the foundation for more advanced channel estimation techniques. For example, Zhou et al. [23] introduced a confidence measure (CM) channel estimation method to distinguish between multi-path and noise components in OFDM systems. They also proposed a novel OFDM channel estimation method based on statistical frames and confidence levels. These studies highlight the importance of incorporating confidence measures into channel estimations. The application of deep learning for channel prediction is relatively new. Bhattacharya et al. [24] discussed the use of an LSTM-based neural network to predict channel characteristics, representing an initial step toward integrating modern machine learning techniques in wireless communications. Our work significantly expands on these previous ideas by developing a multi-task LSTM network that simultaneously predicts multiple channel characteristics along with confidence scores. This multiple-objective approach, which predicts fading, shadowing, attenuation, interference, path loss, and noise along with confidence scores, is novel in this area of research. Our approach extends the concept of confidence measures through incorporating confidence prediction as an integral part of a multi-task deep learning model, which provides a more nuanced understanding of prediction reliability. The integration of traditional techniques with modern deep learning approaches bridges the gap between classical signal processing methods and contemporary machine learning techniques in the context of wireless communications. Implementing confidence prediction in LoRa communications is a novel aspect of our study. Furthermore, integrating confidence prediction with multi-task learning for comprehensive estimation and prediction of channel status in wireless communication systems represents a significant advance in this area.

### 2.5. Adaptive Characteristics of LoRa Communication

LoRa communication exhibits several adaptive characteristics that allow for the optimization of its performance based on channel conditions and application requirements:Spreading factor (SF) adaptation. LoRa technology uses chirp spread spectrum (CSS) modulation with an adjustable SF ranging from 7 to 12. This allows for the dynamic adaptation of data rates and ranges. Higher SFs provide a longer range but lower data rates, while lower SFs offer higher data rates within shorter ranges [25].Bandwidth (BW) adjustment. The LoRa system supports multiple bandwidth options (typically 125, 250, and 500 kHz). Lower bandwidths are used for greater range and receiver sensitivity, whereas higher bandwidths enable faster communications [25].Coding rate (CR) flexibility. Error correction is achieved through adaptive coding rates within the LoRa system. The CR can be controlled by the equation(1)CR=4/(4+n),
where *n* ranges from 1 to 4 [25]. The transmission time increases with the code size.Transmission power control. LoRa devices can change their power (usually between 14 and 20 dBm), enabling trade-offs between energy consumption, SNR, and range requirements [26].Channel selection. Due to the nature of Industry–Science–Medical (ISM) bands, being reserved internationally for purposes other than telecommunications, many LoRa devices can adapt their frequencies to avoid interference. In particular, LoRa technology and ISM bands are very similar. This enables the use of LoRa networks in frequency bands that do not require a specific frequency allocation license, thus fueling the widespread adoption of this technology for IoT applications [12].Data rate adjustment. LoRa’s data rate varies based on the above-listed characteristics. The equation for the LoRa data rate is as follows:(2)$$R_{c}=\frac{SF\times BW}{2\times SF\times CR}$$

These adaptive features allow LoRa systems to optimize their performance for various applications and environmental conditions, balancing factors such as range, data rate, energy consumption, and reliability.

### 2.6. Adjustment LoRa Parameters

LoRa networks offer flexible parameter adjustment capabilities, allowing for the optimization of performance and resource efficiency. To address the need for adaptive solutions in diverse IoT scenarios, recent studies have proposed various methods for dynamic parameter selection and optimization, as follows:Adjusting the SF can help to mitigate interference and noise [27]. Higher SF values increase the SNR and transmission range but also increase the time-on-air (ToA) ratio, which can lead to higher collision probabilities. With an increase in SF, the link budget also increases (i.e., increased values against fading/shadowing losses), causing a higher level of performance during these events. Consequently, the signal is more resistant to these negative factors [28].Changing the CR can introduce forward error correction (FEC), allowing errors due to interference and noise to be detected and remedied [27]. Higher CR values provide better interference protection but increase airtime. In situations where fading and attenuation are significant, higher CR values help to boost the robustness of a transmitted signal [28].Boosting a transmission means providing enough power to reach a gateway, as it can attenuate and fade away [8,29]. Increasing transmission power can also help to mitigate shadow effects in a way that increases the chance of overcoming obstacles [9,26].Through increasing the bandwidth, a system can help to alleviate interference by spreading it over a larger frequency range, thus minimizing the effects of narrow-band interference [9,26]. Note, however, that a larger bandwidth means not only a higher data rate but also greater sensitivity to noise. Tweaking the bandwidth can have an impact on the SNR. Increased bandwidth enhances the noise floor, which can harm the SNR [8,9]. Hence, the SNR can be improved in a noisy environment through decreasing the bandwidth.

When developing our LoRa parameter adjustment strategies, we meticulously considered several key factors to ensure optimal performance. These factors include the impacts of bandwidth on interference and noise sensitivity, the balance between the spreading factor (SF) and signal-to-noise ratio (SNR), the role of the coding rate (CR) in error correction, and the influence of transmission power on overcoming obstacles. Through fine-tuning of these parameters, we aim to enhance the reliability and efficiency of LoRa communications, even in challenging environments. Our approach involves continuous monitoring and adaptive adjustments to maintain robust performance, accounting for both short-term variations and long-term trends in channel conditions. This comprehensive strategy allows us to provide a resilient and efficient LoRa network solution tailored to diverse IoT applications.

## 3. Related Works

Recent research has explored the use of deep learning approaches to improve LoRa channel estimation and signal processing.

Huang and Cai (2024) [5] proposed a joint channel estimation and signal detection structure based on convolutional neural networks (CNN-JCESDs) for LoRa systems. Regarding LoRa networks, they tackled channel fading and interference. An analysis of the CNN-JCESD model [5] demonstrated the great potential of deep learning techniques to significantly enhance the performance of LoRa systems under adverse channel environments. To improve the channel state information (CSI) accuracy by jointly performing channel estimation and signal detection—which may improve the performance against co-SF and inter-SF interference—Huang and Cai proposed a new frame structure with pilot signals. They also used layer normalization for data pre-processing. These techniques contribute to the robustness of the model for Rayleigh block-fading channels under various interference scenarios. The CNN-JCESD [5] model exhibited adaptability to different spreading factors and interference levels. Although effective, this method lacked the ability to capture long-term temporal dependencies, which are crucial in dynamic environments where channel conditions change rapidly over time. Furthermore, the high computational complexity and potential latency in processing batches of symbols may pose challenges for real-time applications, particularly in resource-constrained IoT scenarios. In contrast, our proposed multi-task LSTM approach addresses this limitation of CNN-JCESDs by leveraging the network’s ability to learn and remember long- and short-term patterns in channel behaviors and utilizing the sliding window approach with online incremental learning to enable continuous adaptation under changing channel conditions, yielding reduced latency and computation overhead. As a consequence, the responsiveness and accuracy of the model can be improved over time.

Gu et al. [10] proposed the Channel Occupancy-Aware Resource Allocation (CORA) algorithm, which dynamically adjusts the SF, channel (CH), and transmit power (TP) based on the channel conditions. Using this new method, they developed an algorithm that leverages channel activity detection (CAD) to indicate channel occupancy. To solve the CH/SF assignment problem, CORA uses a multi-agent reinforcement learning approach called actor–attention–critic for MAAC. Historical channel occupancy information is a key innovation in CORA. The global and local occupancy matrices are fused to generate a guideline index for CH/SF assignment using the algorithm. This method enables CORA to make better decisions regarding resource allocation. However, the system state observed by each end-device consists of the delayed channel state information (CSI), distances from the end devices to the gateway, historical channel occupancy rates, and delayed system energy efficiency. The results of simulations and field experiments showed that CORA significantly improved the uplink energy efficiency of the system [10]. Reinforcement learning, such as the MAAC technique used in the CORA algorithm [10], enforces adjustments based on learned policies even if the current state might not necessarily require it. This can lead to continuous fluctuations in the parameters, especially if the system is extremely sensitive to transient changes in network conditions. Such behaviors may result in unnecessary adjustments, thereby affecting the stability of the network and resource consumption. Furthermore, reinforcement learning approaches, such as the MAAC technique used in the CORA algorithm [10], do not inherently provide a measure of uncertainty in their decisions. This limitation can result in similar levels of adjustment being applied across varying degrees of performance degradation, potentially leading to sub-optimal parameter tuning and resource utilization. The prediction confidence measure proposed in our approach could potentially offer more insight into the reliability of the predicted channel states, allowing for more informed decision making. With a prediction confidence measure, the system may be more selective regarding when to take action and may only adjust the parameters when the predictions are highly confident. This could lead to a more stable performance compared with reinforcement learning approaches that are used in CORA, which might be more prone to frequent adjustments based on less-certain information.

Sallum et al. [13] proposed an optimization framework for joint SF and carrier frequency (CF) allocation in LoRa wide-area network (LoRaWAN) deployments. More specifically, this methodology employs mixed-integer linear programming (MILP) to maximize the data extraction rate (DER) and minimize packet collisions and energy consumption. The results showed that the framework led to vastly improved network performance, as evidenced by a 6% average increase in DER and a 13× reduction in collisions, when compared to traditional parameter assignment policies. It also outperformed dynamic radio parameter assignment policies, showing increases in the DER and fewer collisions [13]. Although the MILP approach can satisfy optimal static optimization, the dynamic characteristics of the channel conditions are not considered in real time. This limitation makes it difficult to perform well in environments that change rapidly.

To address the limitations of MILP, our approach performs a gradual adjustment process based on a prediction confidence measure decision. Dynamic MILP can optimize the solution for any set of changed channel conditions. This allows the system to constantly adjust its predictions and adapt its parameters via a sliding window learning process with adaptive thresholding and online incremental learning that reduce training latency and enhancing real-time performance. The predictions from such pipeline augmentations can be more reliable, and the measure of prediction confidence can serve as a measure of the confidence of these predictions, allowing more informed decisions about when and how to adjust the SF and CF allocations to be made. This allows for a constantly adaptive and iterative optimization process that is better suited to the evolving context of LoRaWAN deployments.

Silva et al. [27] presented the SlidingChange mechanism, which is an intelligent method for adaptively tuning the parameters of LoRa networks. Such a mechanism is well suited to our problem as SNR measurements can vary significantly in the short-term, causing unnecessary streaming updates and reconfigurations. The SlidingChange algorithm resulted in significant improvement in network output. Compared with the InstantChange technique, it enhanced the SNR by approximately 37% and lowered the network reconfiguration rate by approximately 16%. The best balance between the SNR gains and the carrier changes was obtained with a sliding window length of 20 units. SlidingChange does not rely on any type of predictive modeling; the decision to change parameters is based on the moving average of SNR values [27]. This averaging approach smooths out any fluctuations but cannot be used to predict future channel changes. To address the limitation of SlidingChange, our approach integrates predictive modeling techniques with the sliding window concept. Through incorporating machine learning algorithms such as LSTM networks, we allow the system to learn patterns in time-series data and make predictions about future channel conditions. This permits more proactive parameter adjustments, potentially further improving the SNR and reducing unnecessary reconfigurations. Additionally, through implementing a prediction confidence measure, we ensure that parameter adjustments are made only when the predictions meet a certain reliability threshold, thus balancing responsiveness with network stability.

The authors of [6] presented a bidirectional LSTM stacked autoencoder fed with a Bayesian surrogate Gaussian process (BSGP-BLSTM-SAE) for LoRaWAN performance analysis and prediction. This model accounts for the sophisticated interaction where ambient weather features interact with in-network metrics such as RSSI and SNR. The BSGP-BLSTM-SAE model achieved high accuracy (approximately 0.45 mean absolute error) in predicting the performance indicators and ambient conditions of the environment. It uses Bayesian optimization, which is used to tune the model hyperparameters and improve the prediction performance. This method establishes a foundation for the real-time optimization of transmission parameters under opportune environmental conditions. The bidirectional LSTM components make the model inherently complex, which implies that the associated computational burden may be large due to the increased number of parameters, the necessity of processing input sequences in both forward and backward directions, and the resultant higher memory and computation requirements. Due to its complexity, with the added latency cost of bidirectional LSTM components, this method might not be a good candidate for real-time monitoring [6]. This model achieves high accuracy at the cost of high computational requirements, which can be difficult to achieve in the context of limited-resource IoT devices. Due to its high accuracy, BSGP-BLSTM-SAE is suitable for applications in which resource constraints are less critical and high reliability is essential. To address the limitations of BSGP-BLSTM-SAE, our proposed approach focuses on developing a lightweight model that maintains high predictive accuracy while reducing computational complexity by allowing for parallel processing and resource sharing. This could involve the use of the multi-task LSTM with online learning algorithms that can adapt to changing conditions with a lower latency. Additionally, incorporating a prediction confidence measure could allow for the selective use of a more complex model only when high-confidence predictions are required, balancing accuracy and computational efficiency. Through combining these techniques with a sliding window approach, real-time performance optimization can be achieved while maintaining the ability to capture complex environmental interactions.

Palacio et al. [8] addressed the energy efficiency challenge in IoT applications by focusing on the LoRaWAN protocol. They developed machine learning models for accurate path loss and shadow fading predictions by incorporating a comprehensive set of environmental variables. This approach involved a wide range of environmental factors, including distance, frequency, temperature, relative humidity, barometric pressure, particulate matter, and SNR. This comprehensive consideration of variables allows for more accurate predictions of network conditions. Various machine learning techniques, including multiple linear regression, support vector regression, random forests, and artificial neural networks, were used to achieve high prediction accuracy. The practical application of these models resulted in substantial energy savings, with a reduction of up to 43% compared to the traditional ADR protocol. Although the study demonstrated significant improvements in energy efficiency, it did not explicitly address real-time adaptation to rapidly changing network conditions. The use of multiple machine learning techniques may introduce complexity in model selection and maintenance for different scenarios. In addition, the study did not mention the computational overhead of running these models on resource-constrained IoT devices. To build upon the work, we focused on developing a unified model that combines the strengths of various machine learning techniques while maintaining computational efficiency. Incorporating online learning mechanisms can allow for real-time adaptation to changing environmental conditions, and implementing a prediction confidence measure can help to dynamically select the most appropriate model or technique based on the current network state and required energy efficiency. This could potentially lead to even greater energy savings while ensuring robust performance across diverse and dynamic IoT environments.

Existing methods, such as CNN-JCESDs, MILP, SlidingChange, CORA, and BSGP-BLSTM-SAE, exhibit limitations, including the inability to adapt continuously under changing channel conditions, high computational complexity, and potential instability due to frequent adjustments based on uncertain predictions. Our proposed multi-task LSTM approach overcomes these challenges by leveraging long- and short-term pattern recognition and a sliding window mechanism with online incremental learning. This enables real-time adaptation and reduces latency while incorporating a prediction confidence measure for selective and reliable parameter adjustments. Multi-task LSTM reduces the computational burden caused by BLSTM by leveraging parameter sharing across tasks. In a multi-task setup, a single LSTM network was trained to estimate multiple states simultaneously rather than having separate models for each task. By capturing shared patterns and dependencies across tasks, multi-task LSTM avoids duplicating the efforts required for the feature extraction. This leads to a more efficient usage of resources while maintaining high prediction accuracy. Gradual adjustment of LoRa parameters guided by a confidence measure strikes a balance between responsiveness and stability. By selectively adjusting parameters based on the channel state rather than applying changes indiscriminately, this approach addresses network issues more effectively and mitigates degradation. These advantages make multi-task LSTMs particularly suited for resource-constrained IoT devices and dynamic environments where real-time adaptability is crucial. By balancing responsiveness, accuracy, and computational efficiency, our approach achieves optimized performance and enhanced energy efficiency, making it well suited for the dynamic environments of LoRaWAN deployments and IoT systems.

Table 1 summarizes the technical aspects of the related studies.

The studies summarized in Table 1 have addressed various aspects of the challenges identified in LoRa network optimization. Although these approaches have made significant strides in this area, they still face limitations in fully addressing the complex and dynamic nature of LoRa networks. Real-time processing and energy efficiency continue to be concerns, particularly in computationally intensive approaches such as the BSGP-BLSTM-SAE model [8]. Various research developments, such as Huang and Cai’s CNN-JCESD [5] and the CORA algorithm [10], partially address the need for comprehensive feature extraction and adaptive parameter optimization. However, these solutions do not fully capture the long-term temporal dependencies which are crucial for stable network performance. Our study builds on these studies by addressing the identified gaps. We propose a novel multi-task approach that combines efficient feature extraction with adaptive parameter adjustment, with the aim of balancing network adaptability and stability while maintaining computational balance. This approach is designed to capture long- and short-term temporal dependencies, manage interference effectively, and optimize multiple LoRa parameters simultaneously based on a prediction confidence measure, thereby addressing the limitations observed in the cited studies.

## 4. Materials and Methods

This section presents a comprehensive methodology to adaptively optimize the LoRa parameters according to the real-time estimate of the channel state. We combine theoretical work on wireless communications with practical considerations to meet the requirements of LoRaWAN systems operating in urban conditions that are subject to highly dynamic environments. The system implements a four-stage architecture, as shown in Figure 1, to optimize the LoRa parameters. The first stage focuses on data acquisition, involving the collection of real-time RSSI, SNR, and configuration parameters from LoRa nodes. The second stage performs feature extraction using a sliding window to capture recent changes in data. The third stage involves estimation of the channel state under different phenomena, including interference, noise, path loss, attenuation, fading, and shadowing. In the fourth stage, LoRa optimization is performed, which involves executing the parameter adaptation logic through implementing a confidence-based decision system that adapts the SF, CR, BW, and TP values according to the detected channel states while maintaining network stability.

### 4.1. Data Collection and Feature Extraction

Accurate estimation of the state of the channel in LoRa networks is essential to optimize communication parameters and ensure reliable long-range transmissions. However, existing LoRa datasets often lack the granularity and labeling required for precise analysis of the channel state and the application of supervised learning techniques. To address these limitations, we propose an adaptive real-time feature extraction methodology that takes advantage of the fundamental characteristics of LoRa signals, including SNR and RSSI, while considering unique LoRa configuration settings.

Our approach extracts both continuous and binary classification features, thereby providing a comprehensive set of indicators for various channel phenomena. The feature extraction process is grounded in well-established communication system equations and is supported by related studies, ensuring its theoretical rigor and empirical validity. This methodology enables accurate characterization and analysis of the communication channel, bridging the gap between raw signal data and actionable insights.

#### 4.1.1. Data Collection

This study uses the LoRa-at-the-edge dataset (LOED) [15], which provides a number of important radio frequency parameters (RSSI, SNR, etc.) that are essential for calculating various indicators of channel conditions and carrying out adaptive control of LoRa communication parameters. Furthermore, the quantitative data also consist of essential LoRa configuration parameters, such as SF, BW, CR, and frequency, which are critical for obtaining a complete understanding of the analytical context of the network.

The dataset captures LoRaWAN traffic in a typical dense urban setting, with a path loss exponent of β=2.4 [30] and 14 dBm transmit power in the 868 MHz ISM band [31], which is the standard for LoRaWAN deployments in Europe. LoRa can operate under very low SNR conditions, which is crucial for long-distance, low-power communications. The SNR limits that ensure reliable signal demodulation are as follows [10,32,33]:(3)$${\mathrm{SNR}}_{\mathrm{limit}}\left(SF\right)=\left\{\begin{array}{cc} -7.5\;\mathrm{dB} & \mathrm{if}\;SF=7\\ -10\;\mathrm{dB} & \mathrm{if}\;SF=8\\ -12.5\;\mathrm{dB} & \mathrm{if}\;SF=9\\ -15\;\mathrm{dB} & \mathrm{if}\;SF=10\\ -17.5\;\mathrm{dB} & \mathrm{if}\;SF=11\\ -20\;\mathrm{dB} & \mathrm{if}\;SF=12 \end{array}\right.$$

#### 4.1.2. Feature Extraction

We developed an advanced feature extraction pipeline to derive meaningful features from the raw RSSI and SNR data. This pipeline employs statistical analysis and domain-specific knowledge to extract both continuous and binary features, thereby enabling a detailed characterization of the channel state. The features extracted are organized and stored as a dictionary structure before being added to the LSTM as input. The dictionary structure acts as a container holding the input information in an accessible and well-structured format. It helps in data pre-processing and ensures that the extracted features are properly aligned and ready for model training. Table 2 presents the considered channel issues and their associated equations.

#### 4.1.3. Adaptive Sliding Window

To capture the long- and short-term variations in the channel conditions, we implemented a sliding window technique. Specifically, we used window sizes of 5 and 50 as $min_{w}$ and $max_{w}$, respectively, that were empirically determined, see Table A1. We implemented a step size of 1, allowing for continuous updating of statistics with each new data point while maintaining a sufficient historical context to detect long-term trends [18]. This approach reduces latency and computational overhead, making it well suited for real-time applications in dynamic environments. Smaller window sizes ensure responsiveness to rapidly changing conditions, whereas the overlapping design preserves the integrity of long-term trends, and the method ensures that no significant data points are missed between consecutive windows. As shown in Algorithm 1, the adaptive sliding window is adjusted based on the variability of RSSI.
**Algorithm 1** Adaptive sliding window for LoRa channel analysis.1:**Input:** Minimum window size $W_{\min}=5$, Maximum window size $W_{\max}=50$2:Step size S=1, Variability threshold α3:**Initialize Window:** 
$W_{\mathrm{current}}\leftarrow W_{\min}$4:**for** each new sample *i* **do**5:    **Update Adaptive Window Size:**6:    Compute variability of RSSI: $\mathrm{Variability}\leftarrow \sigma ({\mathrm{RSSI}}_{i-W_{\mathrm{current}}+1:i})$7:    **if** Variability>α **then**8:        $W_{\mathrm{current}}\leftarrow \min\left(W_{\mathrm{current}}+S,\;W_{\max}\right)$9:    **else**10:        $W_{\mathrm{current}}\leftarrow \max\left(W_{\mathrm{current}}-S,\;W_{\min}\right)$11:    **end if**12:**end for**

#### 4.1.4. Reliable Pattern Filtering

In our approach, we used the unique characteristics of each phenomenon to formulate rules that help to distinguish between different issues. To apply binary classification, we applied filtering to separate reliable patterns from unreliable patterns, and then we used these reliable patterns to formulate adaptive thresholding, as detailed in Algorithm 2.
**Algorithm 2** Reliable pattern filtering.1:**Input:** 
${\mathrm{SNR}}_{\mathrm{limit}},\mathrm{window}\_\mathrm{size}$2:**Maintain:**3:sliding_window←deque([],maxlen=window_size4:last_thresholding←None5:**Calculate Statistics:**6:**if** 
${\mathrm{SNR}}_{i}\geq {\mathrm{SNR}}_{\mathrm{limit}}$ 
**then**7:    is_reliable←True8:    Append current $\Delta P,\Delta \mathrm{RSSI},\hat{\beta }$ to sliding window9:    Compute rolling statistics using sliding window:10:    $\mu_{\Delta P},\;\sigma_{\Delta P}\leftarrow \mathrm{mean}\;\mathrm{and}\;\mathrm{std}\;\mathrm{of}\;\Delta P_{\mathrm{sliding}\_\mathrm{window}}$11:    $\mu_{\Delta \mathrm{RSSI}},\;\sigma_{\Delta \mathrm{RSSI}}\leftarrow \mathrm{mean}\;\mathrm{and}\;\mathrm{std}\;\mathrm{of}\;\Delta {\mathrm{RSSI}}_{\mathrm{sliding}\_\mathrm{window}}$12:    $\mu \hat{\beta },\;\sigma_{\hat{\beta }}\leftarrow \mathrm{mean}\;\mathrm{and}\;\mathrm{std}\;\mathrm{of}\;{\hat{\beta }}_{\mathrm{sliding}\_\mathrm{window}}$13:    $\mu_{PL},\;\sigma_{PL}\leftarrow \mathrm{mean}\;\mathrm{and}\;\mathrm{std}\;\mathrm{of}\;PL_{\mathrm{sliding}\_\mathrm{window}}$14:    Store current thresholding values:15:    $\mathrm{last}\_\mathrm{thresholding}\leftarrow (\mu_{\Delta P},\sigma_{\Delta P},\mu_{\Delta \mathrm{RSSI}},\sigma_{\Delta \mathrm{RSSI}},\mu \hat{\beta },\sigma_{\hat{\beta }},\mu_{PL},\sigma_{PL})$16:**else**17:    is_reliable←False18:    **if** last_thresholding≠None **then**19:        **Use last stored thresholding values:**20:        $(\mu_{\Delta P},\sigma_{\Delta P},\mu_{\Delta \mathrm{RSSI}},\sigma_{\Delta \mathrm{RSSI}},\mu \hat{\beta },\sigma_{\hat{\beta }})\leftarrow \mathrm{last}\_\mathrm{thresholding}$21:    **else**22:        **Skip channel condition checks due to unreliable SNR and no previous reliable pattern**23:    **end if**24:**end if**

#### 4.1.5. Adaptive Thresholding and Binary Classifications Features

Through analyzing continuous features, we can effectively perform binary classification, allowing for accurate identification and the separation of various channel conditions. This methodology ensures a robust and nuanced understanding of the underlying phenomena and facilitates a precise and reliable classification, as shown in Algorithm 3.

In this implementation, we employ 2σ as the threshold to evaluate abnormalities in $\hat{\beta }$, RSSI, PL, and ΔP. This value has been empirically validated to effectively distinguish between reliable and unreliable patterns. Using a threshold lower than 2σ increases sensitivity, which consequently leads to a higher rate of false alerts. In contrast, increasing the threshold beyond 2σ reduces the sensitivity, potentially failing to capture abnormal patterns effectively. The evaluation of noise and interference is based on empirical facts, comparing the power of noise and interference against the signal power. Continuous features offer nuanced insights into gradual changes in channel conditions, while binary classifications enable clear demarcation of significant events, such as fading, shadowing, interference, and others. This dual approach allows for a more complete characterization of the LoRa channel state, bridging the gap between the raw signal data and actionable network insights. Through deriving these features, we not only enhance the resolution of the channel state estimation but also create a pseudo-labeled dataset which is suitable for use with supervised learning algorithms. This innovation addresses the scarcity of labeled data in LoRa research, allowing for the application of advanced machine learning techniques in the context of LoRa network optimization. Thus, our feature extraction process serves as a critical step in transforming limited raw data into a comprehensive set of channel state indicators, thereby advancing the field of LoRa network analysis and adaptive parameter control.
**Algorithm 3** Adaptive thresholding and channel issue classification.1:**Input:** $\mathrm{last}\_\mathrm{thresholding}\leftarrow (\mu_{\Delta P},\sigma_{\Delta P},\mu_{\Delta \mathrm{RSSI}},\sigma_{\Delta \mathrm{RSSI}},\mu \hat{\beta },\sigma_{\hat{\beta }},\mu_{PL},\sigma_{PL})$, $\Delta P_{i}$, $\Delta {\mathrm{RSSI}}_{i}$, ${\hat{\beta }}_{i}$, $PL_{i}$, $P_{{\mathrm{signal}}_{i}}$, $P_{{\mathrm{noise}}_{i}}$, $P_{{\mathrm{interference}}_{i}}$2:**Initialize Flags:**3:is_Fading←False4:is_Shadowing←False5:is_Intereference←False6:is_Noise←False7:is_Attenuation←False8:is_Path_Loss←False9:**Classify Channel Conditions:**10:**if** 
${\hat{\beta }}_{i}>\mu_{\hat{\beta }}+\alpha \cdot \sigma_{\hat{\beta }}\;\mathrm{and}\;\hat{\alpha }<1$ 
**then**11:    is_Fading←True     ▹ Flag as fading if ${\hat{\beta }}_{i}$ exceeds the mean by α standard deviations12:**end if**13:**if** 
${\mathrm{RSSI}}_{i}<\mu_{\mathrm{RSSI}}-2\cdot \sigma_{\mathrm{RSSI}}$ 
**then**14:    is_Shadowing←True     ▹ Flag as shadowing if RSSI is below the mean by 2 standard deviations15:**end if**16:**Noise and Interference Conditions:**17:**if** 
$P_{{\mathrm{noise}}_{i}}>P_{{\mathrm{signal}}_{i}}-P_{{\mathrm{interference}}_{i}}$ 
**then**18:    is_Noise←True     ▹ Flag as noise if noise power exceeds the difference between signal and interference power19:**end if**20:**if** 
$P_{{\mathrm{interference}}_{i}}>P_{{\mathrm{signal}}_{i}}-P_{{\mathrm{noise}}_{i}}$ 
**then**21:    is_Intereference←True     ▹ Flag as interference if interference power exceeds the difference between signal and noise power22:**end if**23:**if** 
$\left|\Delta P_{i}-\mu_{\Delta P}\right|>2\cdot \sigma_{\Delta P}$ 
**then**24:    is_Attenuation←True     ▹ Flag as attenuation if ΔP exceeds 2 standard deviations25:**end if**26:**if** 
$\left|PL-\mu_{PL}\right|>2\cdot \sigma_{PL}$ 
**then**27:    is_Path_Loss←True     ▹ Flag as path loss if PL exceeds 2 standard deviations28:**end if return** Boolean flags indicating detected channel issues

### 4.2. Data Pre-Processing

In the context of LoRa network analysis, robust data pre-processing is crucial for ensuring the quality and reliability of subsequent channel state estimation and parameter optimization processes. Our pre-processing pipeline was designed to handle the complexities of LoRa datasets, which often contain a combination of numeric and categorical data, as well as potential inconsistencies and missing values.

First, we efficiently loaded data from CSV files using Python’s csv module and stored them in a dictionary for easy access. Then, we converted the timestamp strings into date–time objects using a standard format (YYYY−MM−DDHH:MM:SS) to facilitate temporal analysis. Subsequently, automatic feature-type detection categorized columns into numeric and categorical types, thereby reducing manual pre-processing. For numerical columns, we cleaned the data by converting strings to floating values and handling invalid entries with NaN values. Categorical data were encoded using the Scikit-learn LabelEncoder for compatibility with machine learning approaches. We also identified and handled missing data, initially using NaN replacement. Our feature preparation involved scaling with RobustScaler to handle outliers and construct input sequences of length $min_{W}$, capturing the temporal evolution of key features such as RSSI, SNR, SF, and other extracted features, thus enabling LSTM models to learn complex temporal patterns.

This comprehensive pre-processing approach ensured that the LoRa dataset was transformed into a clean, consistent, and machine learning-ready format. Through addressing common data quality issues and standardizing the data structure, we established a solid foundation for advanced feature extraction and channel state estimation techniques. This pre-processing pipeline is crucial for maximizing the value of the LoRa datasets and enabling more accurate and reliable network optimization strategies.

### 4.3. Multi-Task LSTM Model for Channel State Estimation

For the optimization of LoRa networks, accurate estimation of the channel state is paramount for adaptive parameter control. To address this challenge, we propose a sophisticated multi-task LSTM model that simultaneously estimates multiple channel state parameters by leveraging the temporal dependencies inherent in LoRa signal propagation.

Multi-task LSTM architecture: The core of our model is a dual-layer LSTM network with 64 and 32 units, designed to capture short- and long-term dependencies in the signal data, respectively. We incorporated batch normalization and dropout layers (at a rate of 0.3) between the LSTM layers to mitigate over-fitting and improve generalization. This architecture strikes a balance between model complexity and computational efficiency, which is crucial for real-time applications in LoRa networks.Multi-output structure: Our model uniquely outputs predictions for six distinct channel state parameters: fading, shadowing, attenuation, interference, path loss, and noise. Each output utilizes a sigmoidal activation function, enabling the binary classification of significant events in each domain. In addition, we introduced a confidence output that provides a measure of prediction reliability.Loss function and optimization: We employed a carefully weighted multi-task loss function that combines binary cross-entropy for the main outputs and mean-squared error for the confidence estimate. The loss weights were tuned to prioritize critical parameters such as fading (0.5) and shadowing (0.3), reflecting their importance in LoRa communication quality. An Adam optimizer with a learning rate of 0.001 was used, supplemented with the ReduceLROnPlateau callback for adaptive learning rate adjustment.Online incremental learning for real-time adaptation: We applied an online incremental learning scheme [43,44] to allow for real-time implementation in dynamic environments. This is an approach to adapt the model to data without retraining the model end to end, with continuous learning and better performance in non-stationary environments. This process allows for continual improvement of the model over time, adapting to new data instances without forgetting previously acquired patterns.Mathematically, the following equation describes an update rule for the model parameters, which uses stochastic gradient descent (SGD) [43,44] with a regularization component to update the model incrementally as new data arrive, rather than training it from scratch.(4)$$\theta_{t+1}=\theta_{t}-\eta \nabla_{\theta }L\left(\theta_{t},x_{t},y_{t}\right)+\lambda \left(\theta_{t}-\theta_{0}\right)$$All models can be generalized to new channel conditions while still being able to retain information from already learned channel realizations, making them appropriate for real-time applications under progressive LoRa conditions.Model evaluation and confidence-based decision making: Our evaluation methodology incorporates k-fold cross-validation (k = 5) to ensure robust performance assessment. We introduced a novel confidence-based filtering mechanism, in which predictions below the confidence threshold (0.5) are considered unreliable. This approach enhances the practical applicability of the model through providing a clear indicator of prediction certainty. Prediction confidence acts as a measure of reliability for the model’s channel state estimates. High-confidence predictions indicate that the model is more certain about its output, which allows the system to make stronger and more decisive parameter adjustments. Conversely, when the prediction confidence is low (indicating higher mean error), the system avoids aggressive changes to parameters. This prevents overcompensation and unnecessary fluctuations, which might otherwise degrade system performance. Algorithm 4 provides a complete description of the predicted confidence obtained using the proposed approach. We set the confidence threshold for the LSTM as 0.5 because it is a logical midpoint in a binary classification. The confidence score, $Confidence_{i}$, is computed using the sigmoid activation function (σ). Sigmoid outputs values in the range [0, 1]. A threshold of 0.5 naturally divides this range into two parts, making it intuitive to interpret as a decision boundary. Values closer to 1 indicate higher confidence in one class, while values closer to 0 indicate confidence in the other.Adaptive parameter control integration: The model outputs are directly integrated into a decision-making framework for LoRa parameter adjustment. The high-confidence predictions of significant channel state changes trigger specific parameter adaptations, whereas low-confidence predictions prompt minor procedures, as discussed in Section 4.4.

The proposed multi-task LSTM model represents a major advancement in LoRa channel state estimation, offering comprehensive real-time insights into multiple aspects of channel behavior. Through combining sophisticated learning techniques with domain-specific knowledge of LoRa networks, our approach enables more precise and adaptive control of LoRa parameters, ultimately enhancing the network performance and reliability in diverse and challenging environments.
**Algorithm 4** Confidence measure algorithm with MSE.1:**Input:** LSTM hidden state $h_{t}$, weights $W_{c}$, bias $b_{c}$2:**Output:** Confidence score $Confidence_{i}$, Reliability $Reliable_{i}$3:**procedure** 
Confidence Score Computation4:    $Confidence_{i}\leftarrow \sigma \left(W_{c}\cdot h_{t}+b_{c}\right)$5:**end procedure**6:**procedure** 
Threshold Application7:    **if** $Confidence_{i}\geq 0.5$ **then**8:        $Reliable_{i}\leftarrow 1$                        ▹ Prediction is reliable9:    **else**10:        $Reliable_{i}\leftarrow 0$                       ▹ Prediction is unreliable11:    **end if**12:**end procedure**13:**procedure** 
Loss Computation14:    $Loss_{conf}\leftarrow MSE\left(Confidence_{i},Reliable_{i}\right)$15:**end procedure**16:**procedure** 
Reliability Assessment17:    Calculate percentage of reliable predictions18:    Use for model evaluation and monitoring19:**end procedure**

### 4.4. Adaptive LoRa Parameter Algorithm

Based on the outputs of the multi-task LSTM model, including both confidence predictions and detected states, we implemented a graduated response adaptation strategy to adjust the parameters based on the confidence prediction level. This algorithm adjusts the key LoRa parameters in response to the specific channel states identified by the channel state estimation model. This algorithm implements a rule-based approach for adapting LoRa parameters based on the estimated channel state, considering both low and high prediction confidence, as indicated in Algorithm 5. We did not ignore low-confidence predictions entirely, as this could lead to missed opportunities for network optimization and potentially allow channel conditions to deteriorate further. Instead of binary decision making based solely on a confidence threshold, we implemented Mild Adaptation. In Mild Adaptation, minor parameter adjustments were implemented for predictions with confidence levels slightly below the high-confidence threshold; for example, increasing the spreading factor (SF) by only one step (instead of two) or increasing the transmission power (TP) by 1 dBm (instead of 2 dBm).
**Algorithm 5** Dynamic LoRa parameter adjustment.1:**procedure** AdjustLoRaParameters(current_params, channel_state, confidence)2:    **Input:**3:       current_params := {SF,CR,BW,TP}4:       channel_state ∈{noise,interference,fading,attenuation,shadowing}5:       confidence ∈[0,1]6:    **Constants:**7:       SF_MAX := 12, CR_MAX := 0.5, BW_MAX := 500, TP_MAX := 148:       SF_DEFAULT := 7, CR_DEFAULT := 0.8, BW_DEFAULT := 125, TP_DEFAULT := 149:    **function** AdjustParameter(current, step, max_value)10:        **if** current + step ≤ max_value **then return** current + step11:        **elsereturn** max_value12:        **end if**13:    **end function**14:    **function** DefaultParameters15:        **return** {SF_DEFAULT, CR_DEFAULT, BW_DEFAULT, TP_DEFAULT}16:    **end function**17:    **function** AdjustForHighConfidence(SF, CR, BW, TP)18:        **if** channel_state ∈ {noise, interference, fading} **then**19:           SF := AdjustParameter(SF, 2, SF_MAX)20:           CR := AdjustParameter(CR, 0.1, CR_MAX)21:        **end if**22:        **if** channel_state ∈ {noise, interference} **then**23:           BW := AdjustParameter(BW, 125, BW_MAX)24:        **end if**25:        **if** channel_state ∈ {shadowing, attenuation, fading} **then**26:           TP := AdjustParameter(TP, 2, TP_MAX)27:        **end if**28:        **return** {SF, CR, BW, TP}29:    **end function**30:    **function** AdjustForLowConfidence(SF, CR, BW, TP)31:        **if** channel_state ∈ {noise, interference, fading} **then**32:           SF := AdjustParameter(SF, 1, SF_MAX)33:           CR := AdjustParameter(CR, 0.025, CR_MAX)34:        **end if**35:        **if** channel_state ∈ {shadowing, attenuation, fading} **then**36:           TP := AdjustParameter(TP, 1, TP_MAX)37:        **end if**38:        **return** {SF, CR, BW, TP}39:    **end function**40:    **if** channel_state ≠ normal **then**41:        **if** confidence > 0.5 **then**42:           {SF, CR, BW, TP} := AdjustForHighConfidence(SF, CR, BW, TP)43:        **else**44:           {SF, CR, BW, TP} := AdjustForLowConfidence(SF, CR, BW, TP)45:        **end if**46:    **else**47:        {SF, CR, BW, TP} := DefaultParameters()48:    **end if**49:    **if** SF = SF_MAX ∧ CR = CR_MAX ∧ BW = BW_MAX ∧ TP = TP_MAX **then**50:        **for all** parameters ∉ detected issues **do**51:           Reset parameter to default value52:        **end for**53:    **end if**54:    **Return:** {SF, CR, BW, TP}55:**end procedure**

The LoRa parameters were adjusted based on the detected states. The SF increased in response to noise, interference, or fading, improving the signal robustness at the cost of the data rate. The coding rate (CR) was adjusted to provide additional error correction capabilities when required. The bandwidth (BW) was increased to reduce noise and interference, whereas the TP was increased to overcome shadowing and attenuation effects. The algorithm incrementally adjusts the parameters within their valid ranges, allowing for the gradual optimization of network performance. This gradual adjustment based on prediction confidence reduces network fluctuations, which affect resource consumption and network performance. This adaptive approach allows LoRa networks to dynamically optimize their performance in response to changing channel conditions, potentially improving the reliability and energy efficiency of communications. The algorithm design aligns with the flexibility of the LoRa modulation scheme by leveraging its adaptable parameters to maintain optimal performance under diverse environments.

The executed model is a sophisticated optimization algorithm for LoRa parameter adjustment, which considers the variability of the channels used for long-range wireless communications. It makes use of confidence-based decision making to adapt to stricter conditions when channel state estimations are reliable and conservative conditions under uncertainties. As the algorithm is structured in this manner, the parameter updates are guaranteed to remain within valid ranges, thus avoiding overcompensation.

The resource optimization step at the end is particularly noteworthy, as it addresses scenarios in which all parameters have reached their maximum values. In these cases, the algorithm intelligently resets non-critical parameters to their default values, preserving network resources while maintaining optimal performance with respect to the detected channel issues. This is consistent with advanced adaptive control approaches in wireless networks, which are concerned with optimally adjusting the resource usage according to the given communication channels. The algorithm achieves this through adaptively tuning multiple LoRa parameters as a function of various channel states. Thus, it is capable of delivering a holistic solution to maintain optimal network performance in diverse and only partially favorable environments.

## 5. Results

The results presented in this section were used to evaluate the performance and effectiveness of the proposed adaptive sliding window algorithm and multi-task LSTM in dynamic network environments. The impact of the implementation of LoRa parameter adjustment on a resource-constrained environment is also detailed. The analysis focused on key performance metrics, including adaptability, stability, computational efficiency, and reliability, in order to assess the suitability of the proposed approaches for resource-constrained and critical IoT scenarios.

We evaluated the capability of the adaptive sliding window algorithm in a dynamic network environment under different conditions to determine how well the window sizes can be adjusted dynamically, how well it keeps the system stable, and the ease of response. We evaluated the ability of the adaptive sliding windows to capture long- and short-term variations. Important metrics, including the average window size, the number of adjustments made, and the amount of time spent above the variability threshold, were examined to assess the trade-off between flexibility and stability achieved by the algorithm.

In addition, the multi-task LSTM approach was evaluated in terms of its prediction, accuracy, recall, and F1-score. An experimental comparison was performed with other state-of-the-art approaches, including CORA, CNN-JCESD, ML-ADR, and BSGP-BLSTM-SAE, in order to demonstrate the better overall balanced performance of the proposed method in terms of reliability, resource efficiency, and adaptability.

The findings presented in this section allow for a comprehensive understanding of the capabilities of the proposed methods, emphasizing their potential to improve the efficiency, reliability, and adaptability of communications in challenging IoT environments.

### 5.1. Performance of Adaptive Sliding Window

The performance of the adaptive sliding window algorithm was evaluated to assess its effectiveness in dynamic network environments. The algorithm was implemented with the following parameters: W_min = 5, W_max = 50, and step size = 1, where the initial window size was W_min. An analysis of 200 test samples yielded the following key results. The average window size in all samples was 20.38 and the median window size was 16.0, indicating that the algorithm predominantly maintained smaller window sizes to enhance its responsiveness and reduce computational overhead and resource consumption, as shown in Figure 2.

A key finding is that the algorithm dynamically adapted the window size, making 186 adjustments during the observation period. The number of changes was generally balanced, with 102 increases and 84 decreases, indicating the adaptability of the algorithm to changing network conditions. The adaptation of the step size (S=1), in which the increase and decrease were limited to ±1, allowed for gradual changes, keeping the system stable by avoiding rapid transitions.

The algorithm spent 53.77% of the time above the variability threshold (α=7.762), suggesting that it responded appropriately to the varying signal while simultaneously maintaining stability.

As shown in Figure 2, the distribution indicates that the algorithm has achieved a good balance between stability and flexibility. The moderate average window size of 20.38 indicates that the algorithm struck a balance between $W_{m}in$ and $W_{m}ax$, achieving stability during low-variability periods and improving adaptability during high-variability periods.

This distribution pattern indicates that the algorithm effectively balanced maintaining responsiveness (smaller windows) while still allowing for adaptation to changing network conditions when necessary. This behavior provides some important advantages. A lower window size reduces computational overhead and minimizes processing requirements for real-time applications. It also reduces latency in the network response while improving adaptability to rapid changes in network conditions. This approach can also be adapted to detect long-term variations, such as shadowing, through analyzing the variability and trends in RSSI values over maximum window sizes, as shown in Figure 3.

### 5.2. Performance Evaluation of Feature Extraction Method and Adaptive Thresholding Technique

Based on our feature extraction method—which analyzes the interference power, noise power, path loss, attenuation level, and gamma parameters—we observed the following parameter ranges in the LOED:Noise power: The measurements range from −115 dB to −80 dB, which are in accordance with typical LoRa noise floor measurements in the literature [45,46].Interference level: The values range from 0 to 22 dB, providing a comprehensive range for interference detection in LoRa networks [47].Attenuation level: The observed range of 70.18 to 141.64 dB corresponds to the expected LoRa link budgets, which typically accommodate attenuation up to 140–150 dB in urban environments [48].Path loss: The measurements of 124.84 to 205.84 dB align with theoretical path loss models for LoRa communications, which predict values between 120 and 200 dB depending on distance and environmental factors [48].The scale parameter (signal variance): The analysis revealed variations encompassing the range of 8 to 18 dB, while $\delta_{r}ssi$ provide variations between −25 and −5 db, which are consistent with established parameters for multi-path fading detection in LoRa communications [48].The shape parameter: This parameter determines the shape of the gamma distribution, affecting the skewness (asymmetry) and the concentration of the distribution. It ranges from 0.7 to 1.3 and takes three main values:(5)$$\hat{\alpha }\left\{\begin{array}{cc} \hat{\alpha }<1 & \mathrm{The}\;\mathrm{distribution}\;\mathrm{is}\;\mathrm{heavily}\;\mathrm{skewed}\;\mathrm{to}\;\mathrm{the}\;\mathrm{lower}\;\mathrm{values}.\\ \hat{\alpha }=1 & \mathrm{The}\;\mathrm{distribution}\;\mathrm{becomes}\;\mathrm{an}\;\mathrm{exponential}\;\mathrm{distribution}.\\ \hat{\alpha }>1 & \mathrm{The}\;\mathrm{distribution}\;\mathrm{resembles}\;a\;\mathrm{normal}\;\mathrm{distribution}\;\mathrm{for}\;\mathrm{larger}\;\mathrm{values}\;\mathrm{of}\;\hat{\alpha }. \end{array}\right.$$

The observed ranges validate our feature extraction methodology, as they consistently fall within both the theoretical predictions and empirical ranges established in previous LoRa studies. These values demonstrate a strong correlation with urban and suburban deployment scenarios, where path loss and attenuation effects are significant, yet remain within LoRa’s operational parameters.

Regarding adaptive thresholding, our method effectively separates reliable communication patterns from unreliable ones by comparing parameters against an adaptive dynamic threshold derived from reliable patterns. This approach successfully detected 87.21% of reliable communications as normal patterns and 82.81% of unreliable communications with various issues as abnormal patterns. Table 3 summarizes the results obtained with the binary classification and thresholding techniques.

An analysis of the first 200 samples revealed various communication issues that affected network performance. For short-term issues, significant attenuation was most prevalent, affecting 24.50% (49 samples), followed by significant path loss, affecting 19.50% (39 samples), as shown in Figure 4.

Fading was detected in 18.00% (36 samples), as shown in Figure 5, which represents the resulting shape and scale parameters used to detect fading.

Signal interference was detected in 15.00% of cases (30 samples), while noise effects were less common but still noticeable (14.50%; 29 samples), as shown in Figure 6.

These findings highlight that environmental and physical factors significantly impact network reliability, with attenuation and noise being the main concerns. For long-term issues, such as shadowing, we implemented a window size of 50 to detect long-term variations. Our analysis identified 49 long-term shadowing events, affecting 24.50% of the samples, as detailed in Figure 7.

### 5.3. Performance Evaluation of Channel State Estimation

We evaluated the performance of the multi-task channel state estimation model to predict fading, shadowing, interference, noise, path loss, and attenuation using a five-fold cross-validation method. Table 4 presents the resulting training and validation performance metrics, allowing for comprehensive understanding of the behavior of the model.

The training and validation losses across all folds were consistently low, in the vicinity of 0.076–0.097. This result indicates that the model learned to simultaneously predict multiple channel states, which is desirable in a multi-task channel state estimation model for LoRa networks. The small gap between training and validation losses indicates excellent generalization ability, which is crucial for reliable performance in real-world LoRa deployments. The training and validation accuracies were remarkably high and consistent across all folds (approximately 97.6% to 97.9%, respectively). This high accuracy is consistent with the detailed performance metrics shown in the cross-validation results, further confirming the effectiveness of the model in estimating the various channel states. The consistency among the folds indicated that the model is robust to variations in the training data, which is essential for reliable channel state estimation in dynamic LoRa environments. The consistency between the training and validation metrics (both loss and accuracy) indicated that the model did not over-fit the training data. The avoidance of over-fitting is vital for channel state estimation models, as it implies that the model will perform well on unseen data in real LoRa network deployments. The generalization ability is particularly crucial, given the diverse and dynamic nature of the wireless channel conditions in LoRa networks. The relatively small variations in the loss and accuracy between the folds indicate a stable learning process. This stability is beneficial for implementing adaptive parameter selection in LoRa networks, as the predictions of the model would then be consistent and reliable over time. Stable predictions are essential for adjusting spreading factors, coding rates, and other LoRa parameters in response to changing channel conditions.

The five-fold cross-validation results, as detailed in Table 5 and Table 6, demonstrate high accuracy and consistency across all channel state predictions, indicating the robustness of the multi-task channel state estimation model. These results are consistent with the objective of developing a reliable estimation system for LoRa networks.

Fading detection revealed a strong performance, with accuracies ranging from 97.93% to 98.21%. The model exhibited high precision and recall for both classes, indicating effective discrimination between fading and non-fading conditions. This accurate fading detection can considerably improve the reliability of LoRa communications.

Shadow detection also achieved excellent accuracy (99.13% to 99.52%), with high precision and recall for both classes. This result indicates the ability of the model to effectively capture the log-normal variance for shadowing detection, which is crucial for elucidating the signal propagation mechanism in LoRa networks.

Interference detection demonstrated strong and balanced performance across both classes, with precision ranging from 97.70% to 98.46%. This balanced detection of interference is crucial for adaptive parameter selection in LoRa networks, particularly in environments in which multiple systems share spectra.

Noise detection revealed consistently high performance across all folds (97.77% to 98.08% accuracy). The balanced precision and recall for both classes indicated effective noise estimation, which is essential for accurate SNR calculations and subsequent parameter adjustments.

Although the multi-task LSTM model achieved notable success in detecting most phenomena, its performance in detecting path loss and attenuation was hindered by an imbalanced dataset. Specifically, for Class 1, representing cases where attenuation or path loss is undetected, the model recorded an F1-score and a recall value of 0.0, indicating a complete failure to identify such instances. This raises a critical concern, since in real-world applications, path loss and attenuation might occur more frequently. Addressing this imbalance is essential for improving the model’s robustness, and future work should focus on incorporating techniques such as data augmentation or resampling to rectify this issue and ensure reliable detection.

We conducted 3-fold validation to evaluate the performance and robustness of our model across different test folds. The results demonstrated convergence with those obtained using 5-fold validation, highlighting the model’s reliability and consistency across various k-fold validation techniques. However, a notable degradation in F1-score and recall was observed in scenarios involving attenuation and path loss. This can be attributed to class imbalance within the dataset, which emphasizes the need for addressing this imbalance to improve model performance in these specific cases. The result is shown in Table A2 and Table A3.

### 5.4. Prediction Confidence Analysis

We evaluated the reliability of the prediction confidence measure for each channel state parameter, and the correlation between prediction confidence and estimation error was plotted for each channel state parameter.

Visualization of prediction confidence against estimation error for each channel state parameter, as shown in Figure 8, provides further insight into the reliability of the multi-task channel state estimation model. This analysis is consistent with the objective of incorporating a prediction confidence measure to improve the effectiveness of parameter adjustments in LoRa networks. The model demonstrated exceptionally high accuracy across all channel state parameters, with accuracies ranging from 97.06% to 99.71%, as shown in Table 7. This result indicates robust performance in terms of the estimation of various channel conditions, which is crucial for the effective optimization of LoRa parameters. Path loss and attenuation revealed particularly tight confidence error relationships, consistent with their higher accuracy rates. Scatter plots and trend lines that illustrate the relationship between prediction confidence and estimation error are particularly valuable. A strong negative correlation between confidence and error indicates that high-confidence predictions are probably more accurate. This information can be used for more reliable parameter adjustments through implementing a confidence threshold to trigger changes in LoRa transmission settings. The high accuracy and confidence-based error analysis indicate that this model can considerably enhance the selection of adaptive parameters in LoRa networks. Considering both the predicted channel state and the associated confidence, a LoRa system can make informed decisions about when and how to adjust the spreading factors, coding rates, and transmission power.

### 5.5. Comparison of Adaptive Mechanisms

We performed a comparative analysis using adaptive algorithms from the existing literature, simulating different approaches using Python 3.10 tools. Using 1000 samples, a comparative analysis was performed using different LoRa network adjustment approaches in our simulation. A total of 100 LoRa nodes, eight channels, and spreading factors between 7 and 12 were taken as the parameters of the simulation setup. Under high energy per packet (fixed at 0.05 Joules) and SNR ranging from -20 to 5 dB, the network performance was evaluated for a maximum distance of 1000 m over 10 epochs. Furthermore, every method followed a sequence of decisions to fine-tune the corresponding LoRa parameters based on each considered approach, allowing for the calculation of important performance metrics such as the packet delivery ratio (PDR), energy consumption, number of operations, and processing time, as presented in Table 8. This exhaustive analysis allowed us to study the trade-offs between energy efficiency, computational complexity, and real-time processing capabilities, as well as to determine the suitability of each approach for use in constrained IoT environments. Through analyzing these metrics, it is possible to identify methods that perform well in specific aspects, as well as their downsides. This can help decision-makers to determine which approach is better suited to their needs, through analyzing the pros and cons of various approaches.

The PDR metric represents the degree of success of the packet delivery rate provided by the model through the network. As can be seen from the table, the multi-task LSTM achieved a significantly higher PDR of 100% (all packets are delivered perfectly), whereas that for SlidingChange was 47%, which is too low for use in real scenarios. The multi-task LSTM obtained such a perfect result as it is equipped with a confidence measure, which is used to decide whether some adjustments in the network are important. It also minimizes frequent volatility by avoiding unnecessary or random parameter changes and ensuring relative constancy. Another reason is that the multi-task LSTM extracts a variety of related features specific to the issue. These features help to predict channel states accurately and subsequently apply gradual adjustments to parameters based on the estimated states. This proactive solution helps to improve channel conditions before they completely degrade based on the result of the confidence measure.

Energy efficiency is of great importance in low-power networks such as LoRa, and using less energy creates a sustainable and long-lived network. The multi-task LSTM again performed quite well, with an energy consumption of approximately 0.07987 joules; in comparison, CORA consumed up to 50 joules per epoch. Another metric is the processing time, which is proportional to the time required for an algorithm to process or update network parameters. For real-time adjustments, shorter processing times are preferred. The multi-task LSTM achieved a processing time of approximately 0.00016 s, which was very fast when compared to the other approaches. The multi-task LSTM model excelled in comparison with other models mentioned in the literature based on energy consumption and processing time, as the proposed model possesses the ability to share parameters and computations for multiple tasks, leading to a significant reduction in the number of parameters to be trained. This alleviates the computational burden and the energy required for training. Another reason is that through training multiple tasks simultaneously, the model can improve its generalization performance. This can result in improved overall accuracy, as well as faster network convergence during training. Furthermore, incorporating online incremental learning into the multi-task LSTM model reduces the training overhead and allocates resources more efficiently by updating the model incrementally, focusing on the relevant data changes rather than the entire dataset. With regard to the computational complexity, this metric measures the amount of computation (operations) performed by each scheme, which is useful for determining how computationally expensive a method is. Note that multi-task LSTM required a much higher number of operations (17,216,000 ops) than CORA (100,000 ops) or SlidingChange (∼10 ops); therefore, although some methods can be energy-efficient and yield reliable PDR, they may require more processing resources.

The multi-task LSTM model demonstrated flexibility and a high prediction accuracy of 96.3%, allowing it to achieve efficient resource management and performance. The high PDR, combined with adaptability and increased efficiency, indicates that the multi-task LSTM is well suited for use in performance-centric and resource-constrained environments.

The ML-ADR is a noteworthy alternative that balances reliability with efficiency, making it highly suitable for resource-constrained IoT environments. It achieved a high packet delivery ratio (PDR) of 99% while maintaining a moderate energy consumption of 0.500 Joules per packet. Its processing time of 0.0310 s ensures that adjustments are made promptly, which is critical in dynamic and real-time network settings. Although its computational complexity (approximately $2.24\times 10^{6}$ operations) is higher than that of some simpler methods, it remains manageable for centralized or edge processing. In general, ML-ADR excels in applications where maintaining near-optimal network performance with limited energy resources and rapid decision making are paramount, making it a robust solution for constrained IoT deployments.

Similarly, approaches such as BSGP-BLSTM-SAE, despite their high PDR, were found to have significant computational demands and consume significant processing time.

This analysis demonstrated that while some approaches excel in individual metrics, the proposed multi-task LSTM achieved the best overall balance. Its adaptive algorithm maintains high reliability (100% PDR) and reasonable resource utilization, making it suitable for dynamic IoT environments where both performance and resource efficiency are crucial. Stable scalability metrics, as evidenced by its consistent performance under various conditions, confirm that the proposed network maintains its operational capacity despite challenging channel conditions. This stability is achieved through an intelligent parameter adjustment mechanism that proactively responds to changing channel states, in contrast to conventional approaches which only reactively address issues after they occur.

## 6. Conclusions

In LoRa networks, the proposed multi-task LSTM model enables improved performance in channel state estimation and promotes the adaptation of LoRa parameters to changes in IoT environments. We applied the adaptive sliding window technique with a step size of 1 to extract detailed features, which helped to better estimate the channel state. This holistic view can be helpful in understanding the behaviors of wireless channels, which is necessary to achieve adaptability in control strategies for LoRa networks and high accuracy in estimating long- and short-term multi-channel impairments. The adaptive sliding window algorithm was found to be capable of striking a good balance between adaptability (in the case of large signal variations) and stability (in the case of small signal variations), mainly maintaining small windows (average size of 20.38). Therefore, it promotes computational resource savings and low latency. This fits the requirements for use in LoRa networks, namely, being able to estimate fading, shadowing, noise, and interference in real time. In addition, attenuation and path loss events can be estimated with more attention to power conservation and real-time adaption. The use of the proposed methods allows for the successful detection and quantification of various channel impairments, particularly using a dual classification method consisting of rule-based and LSTM approaches.

The rule-based method can be used to generate labels or perform the initial classification of unlabeled data, comparing each instance with an adaptive threshold that is computed considering a reliable pattern.

The LSTM method can then facilitate a more nuanced and deeper analysis, potentially benefiting from the labels generated by the rule-based method. As can be seen from the results, the accuracy of the multi-task LSTM model remained high (97.6–97.9%) for all channel states, implying its good performance in estimating diverse channel conditions. This is consistent with the objective of creating a reliable estimation system for LoRa networks. The proposed model can effectively estimate the states of different channels simultaneously, which is very important for optimizing LoRa networks, and its good reliability—in terms of precisely discovering fading, shadowing, and interference—can improve the robustness of communications in LoRa networks. This enables the model to make better decisions regarding the adjustment of spreading factors, coding rates, and transmission power, depending on the evolution of the channel. The implementation of a confidence measure as another output from the LSTM model provides insight about the level of channel state uncertainty, thus helping to apply gradual adaption based on the level of confidence, subsequently conserving resources and avoiding unnecessary adaption.

Empirically, the multi-task LSTM was simulated using Python tools, and the following results were achieved. A packet delivery ratio of 100% was obtained using the multi-task LSTM approach, which was better than those of the other methods, while maintaining efficiency in resource consumption. Hence, both its performance and its efficiency make it ideal for use in dynamic IoT environments. Although the multi-task LSTM did not achieve the best computational complexity, it shared most of its operations among six tasks, providing balanced performance and resource management and, thus, making the model suitable for IoT deployments. The robustness of the model under different conditions suggests its good scalability and an ability to maintain operational capacity despite challenging channel conditions. Although the multi-task LSTM model performed well in detecting most phenomena, the detection of path loss and attenuation was compromised due to the imbalance of the dataset. In particular, for Class 1 (representing cases where attenuation or path loss is not detected), the model recorded zero for both the F1-score and recall. A relevant concern is that in real-world scenarios, path loss and attenuation might be more prevalent; as such, this issue should be resolved in future work. Our future research will emphasize the practical implementation of the proposed model in real-world applications that demand reliability and adaptability, such as healthcare monitoring, environmental tracking, and smart city solutions. Using the adaptive capabilities of the model in real time, our aim is to address the unique challenges posed by these critical domains, ensuring robust performance and efficient resource management in dynamic scenarios. In general, the proposed multi-task LSTM model, combined with an adaptive estimator, provides an effective method to improve the key properties of LoRa communications in difficult IoT scenarios.

## Figures and Tables

**Figure 1 sensors-25-02121-f001:**
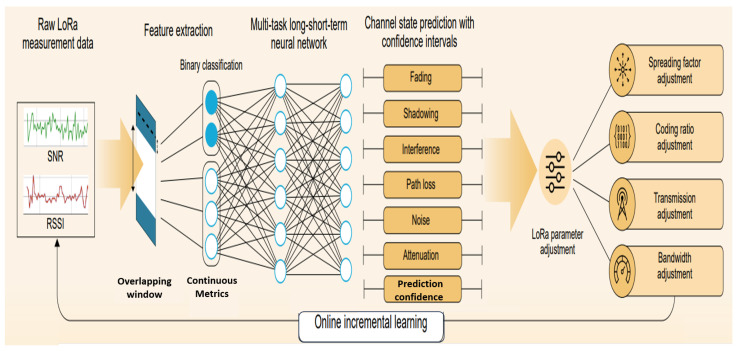
Multi-task LSTM architecture.

**Figure 2 sensors-25-02121-f002:**
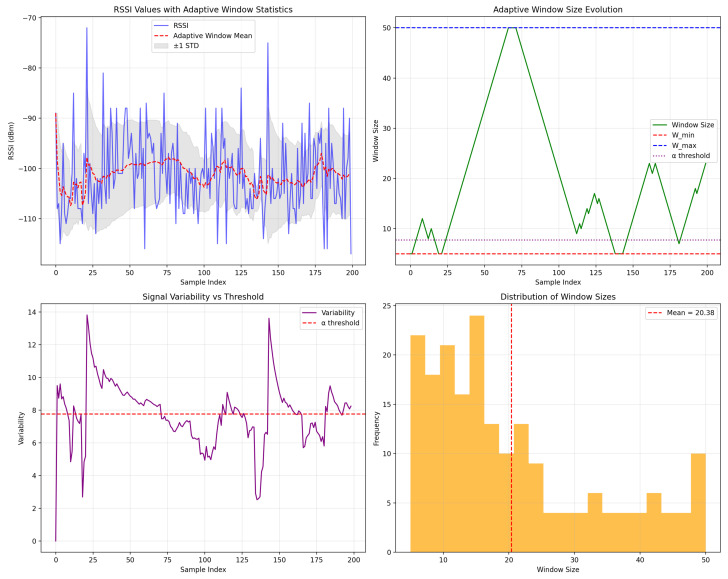
Performance evaluation of adaptive sliding window algorithm.

**Figure 3 sensors-25-02121-f003:**
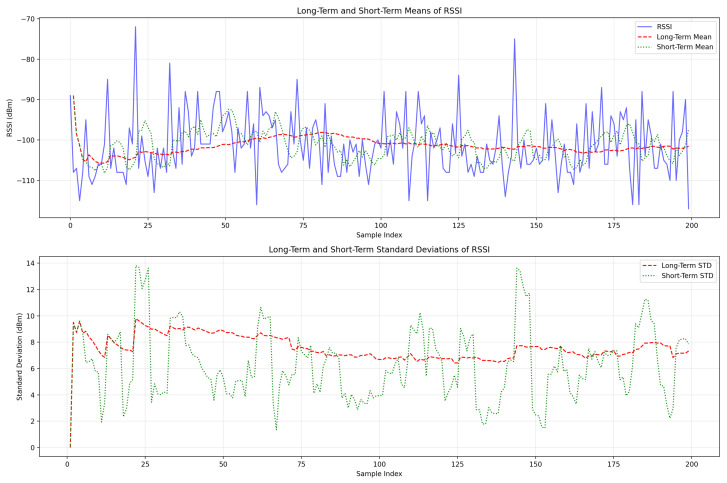
Adaption of window size to detect long-term and short-term variations.

**Figure 4 sensors-25-02121-f004:**
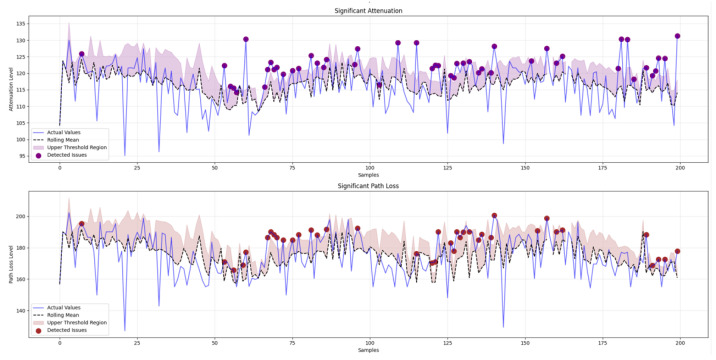
Attenuation and path loss analysis results for the first 200 samples.

**Figure 5 sensors-25-02121-f005:**
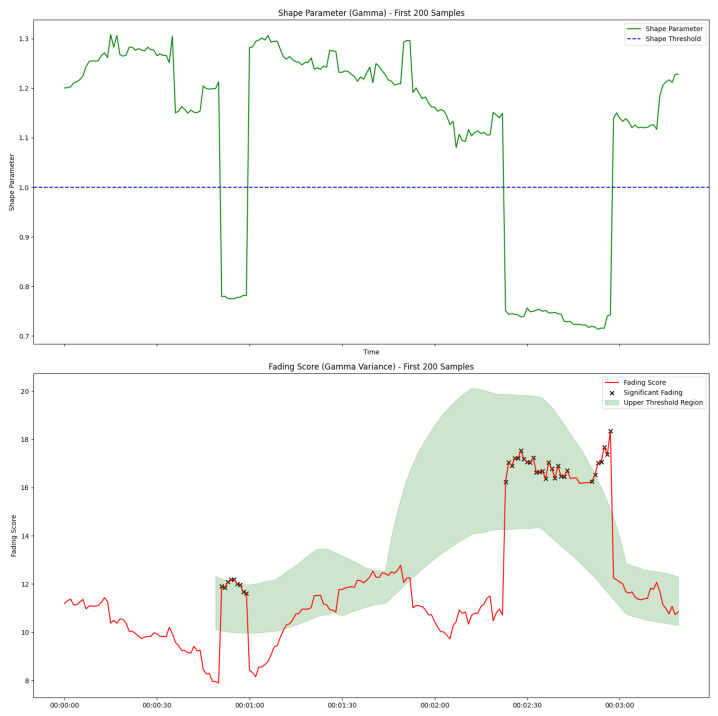
Fading analysis results for the first 200 samples.

**Figure 6 sensors-25-02121-f006:**
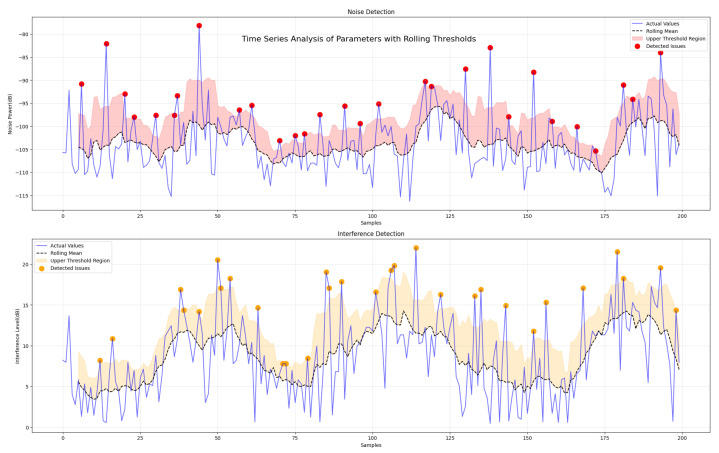
Interference and noise analysis result for the first 200 samples.

**Figure 7 sensors-25-02121-f007:**
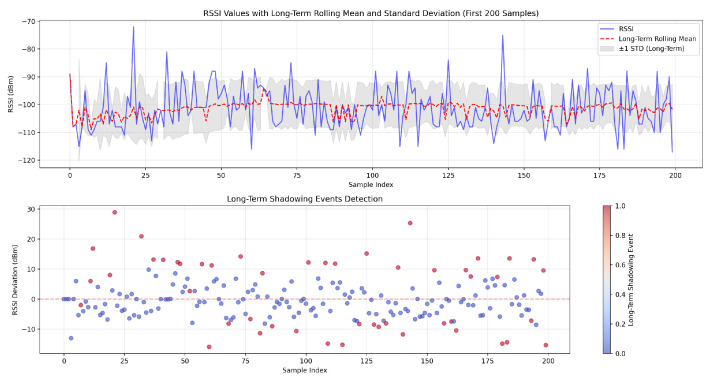
Shadowing analysis results for the first 200 samples.

**Figure 8 sensors-25-02121-f008:**
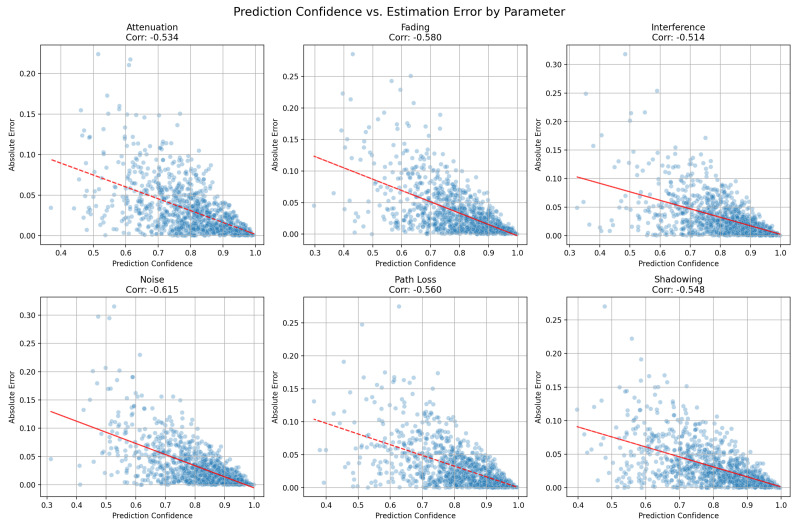
Prediction confidence versus estimation error for each channel state parameter.

**Table 1 sensors-25-02121-t001:** Comparative analysis of LoRaWAN network optimization studies.

Study	Features	Target States	Extraction Methods	Issues Detected	Adjusted Parameters
CORA algorithm [10]	Channel gains, distances, historical channel occupancy rates, system energy efficiency, SNR	Optimal SF and CH assignment, TP allocation	Domain knowledge: CAD; statistical method: matrix-merging	Channel congestion, sub-optimal power allocation	SF, channel, transmit power
SlidingChange mechanism [27]	SNR, SF, BW, carrier frequency	Optimal SF, BW, and F configuration	Statistical method: sliding window, domain knowledge	SNR fluctuations, bandwidth inefficiencies	SF, BW, frequency
ML-ADR [8]	Path loss, distance, frequency, temperature, humidity, pressure, particulate matter	Path loss prediction	Pre-processing: Mahalanobis distance, environmental factors	Inaccurate path loss predictions	Transmission power, environmental adjustments
CNN-JCESD [5]	Dechirping, DFT, real and imaginary parts, layer normalization	Channel fading and interference mitigation	Frame structure, Rayleigh block-fading model, layer normalization	Channel fading, co-SF and inter-SF interference	Frame structure, SNR range
SGP-BLSTM-SAE [6]	RSSI, SNR, ambient weather conditions	Channel performance prediction, ambient condition estimation	BSGP-BLSTM_SAE, Bayesian optimization	Weather–network performance interplay	Transmission parameters, hyperparameters
LR-CPLS [13]	SF, CF, data extraction rate	Maximize DER, reduce collisions and energy consumption	MILP, approximation algorithm	Packet collisions, computational complexity	SF, carrier frequency

Note: SF = spreading factor, CF = carrier frequency, BW = bandwidth, SNR = signal-to-noise ratio, RSSI = received signal strength indicator, DER = data extraction rate, MILP = mixed integer linear programming, BSGP-BLSTM-SAE = Bayesian surrogate Gaussian process-based bidirectional LSTM stacked autoencoder.

**Table 2 sensors-25-02121-t002:** Channel issues and their associated equations.

Issue Type	Description	Equations
Fading	Rapid fluctuations due to multi-path propagation over short-term intervals. Detection involves monitoring short-term signal strength variations using Gamma parameters for scale and shape [34,35].	$\hat{\alpha }=\frac{\mu_{\mathrm{RSSI}}^{2}}{\sigma^{2}}$ and $\hat{\beta }=\frac{\sigma^{2}}{\mu_{\mathrm{RSSI}}}$
Noise power	Unwanted signals interfering with data transmission and reception [36].	$\mathrm{signal}\_\mathrm{power}=\mathrm{RSSI}+\mathrm{SNR}-10\log_{10}\left(1+10^{\frac{\mathrm{SNR}}{10}}\right)$ and $\mathrm{noise}\_\mathrm{power}=\mathrm{RSSI}-10\log_{10}\left(1+10^{\frac{\mathrm{SNR}}{10}}\right)$
Interference	Signal degradation caused by other devices [36,37].	$P_{\mathrm{interference}\_\mathrm{dB}}=\mathrm{RSSI}-P_{\mathrm{signal}\_\mathrm{dB}}$
Shadowing	Slow signal variations due to obstacles which can be detected using log-normal variance [38,39,40].	$\mathrm{RSSI}\_\mathrm{linear}=10^{\frac{\mathrm{RSSI}}{10}}$ and $\log\_\mathrm{variance}=\mathrm{Var}\left(\log(\mathrm{RSSI}\_\mathrm{linear})\right)$
Attenuation	Reduction in signal strength [41].	$\Delta P=TP-P_{\mathrm{signal}}$
Log-distance path loss	Empirical model accounting for distance and environmental factors [8,42].	$PL=PL_{0}+10\times \beta \times \log_{10}\left(\frac{d}{d_{0}}\right)$ where $d_{0}=10^{\frac{{\mathrm{RSSI}}_{0}-\mathrm{RSSI}}{10\cdot \beta }},$ $PL_{0}=20\cdot \log_{10}\left(f\right)+20\cdot \log_{10}\left(\frac{4\pi }{c}\right),$ $d=10^{\frac{L_{\mathrm{FSPL}}-20\log_{10}\left(f\right)-20\log_{10}\left(\frac{4\pi }{c}\right)}{20}}$ and $L_{\mathrm{FSPL}}=TP-\mathrm{RSSI}$

**Table 3 sensors-25-02121-t003:** Performance metrics for different spreading factors (SF) in LoRa communications.

SF	Accuracy (%)	Precision (%)	Recall (%)	F1 (%)	Sample Size
7.0	78.88	82.56	88.76	85.55	11,015.0
8.0	81.98	84.31	93.26	88.56	3752.0
9.0	70.13	68.31	86.64	76.39	1945.0
10.0	72.17	69.08	86.67	76.88	309.0
11.0	76.17	80.00	90.91	85.11	235.0
12.0	90.93	96.48	93.75	95.10	9332.0

**Table 4 sensors-25-02121-t004:** Training and validation results.

Fold	Train Loss	Val Loss	Train Accuracy	Val Accuracy
1	0.0759	0.0926	0.9796	0.9781
2	0.0892	0.0919	0.9765	0.9784
3	0.0873	0.0871	0.9791	0.9781
4	0.0883	0.0937	0.9778	0.9774
5	0.0871	0.0967	0.9771	0.9760

**Table 5 sensors-25-02121-t005:** Five-fold validation results for all states.

Fold	Class	Precision	Recall	F1-Score	Support	Accuracy
Fading						
1	0	0.9896	0.9878	0.9887	4832	0.9793
	1	0.8647	0.8829	0.8737	427	
2	0	0.9911	0.9894	0.9903	4828	0.9821
	1	0.8836	0.9000	0.8917	430	
3	0	0.9897	0.9878	0.9887	4848	0.9793
	1	0.8592	0.8780	0.8685	410	
4	0	0.9938	0.9859	0.9898	4879	0.9812
	1	0.8349	0.9208	0.8758	379	
5	0	0.9903	0.9903	0.9903	4855	0.9821
	1	0.8834	0.8834	0.8834	403	
Shadowing						
1	0	0.9943	0.9967	0.9955	5111	0.9913
	1	0.8750	0.8041	0.8380	148	
2	0	0.9965	0.9982	0.9974	5116	0.9949
	1	0.9323	0.8732	0.9018	142	
3	0	0.9963	0.9971	0.9967	5116	0.9935
	1	0.8913	0.8662	0.8786	142	
4	0	0.9971	0.9980	0.9976	5125	0.9952
	1	0.9219	0.8872	0.9042	133	
5	0	0.9963	0.9977	0.9970	5120	0.9941
	1	0.9084	0.8623	0.8848	138	

**Table 6 sensors-25-02121-t006:** Five-fold validation results for all states.

Fold	Class	Precision	Recall	F1-Score	Support	Accuracy
Interference						
1	0	0.9728	0.9888	0.9807	3115	0.9770
	1	0.9833	0.9599	0.9714	2144	
2	0	0.9839	0.9842	0.9841	3169	0.9808
	1	0.9761	0.9756	0.9758	2089	
3	0	0.9818	0.9910	0.9864	3104	0.9838
	1	0.9868	0.9735	0.9801	2154	
4	0	0.9854	0.9854	0.9854	3157	0.9825
	1	0.9781	0.9781	0.9781	2101	
5	0	0.9844	0.9897	0.9871	3121	0.9846
	1	0.9849	0.9771	0.9810	2137	
Path Loss						
1	0	0.9952	0.9998	0.9975	5234	0.9952
	1	1.0000	0.0000	0.0000	24	
2	0	0.9941	1.0000	0.9970	5227	0.9941
	1	1.0000	0.0000	0.0000	31	
3	0	0.9956	1.0000	0.9978	5235	0.9956
	1	1.0000	0.0000	0.0000	23	
4	0	0.9952	1.0000	0.9976	5233	0.9952
	1	1.0000	0.0000	0.0000	25	
5	0	0.9968	1.0000	0.9984	5241	0.9968
	1	1.0000	0.0000	0.0000	17	
Noise						
1	0	0.9737	0.9724	0.9730	2171	0.9777
	1	0.9806	0.9815	0.9811	3087	
2	0	0.9737	0.9724	0.9730	2171	0.9777
	1	0.9806	0.9815	0.9811	3087	
3	0	0.9795	0.9725	0.9760	2112	0.9808
	1	0.9817	0.9863	0.9840	3146	
4	0	0.9746	0.9765	0.9756	2126	0.9802
	1	0.9840	0.9828	0.9834	3132	
5	0	0.9758	0.9745	0.9751	2154	0.9797
	1	0.9823	0.9832	0.9828	3104	
Attenuation						
1	0	0.9960	1.0000	0.9980	5238	0.9960
	1	1.0000	0.0000	0.0000	21	
2	0	0.9951	1.0000	0.9975	5232	0.9951
	1	1.0000	0.0000	0.0000	26	
3	0	0.9949	1.0000	0.9974	5231	0.9949
	1	1.0000	0.0000	0.0000	27	
4	0	0.9947	1.0000	0.9973	5230	0.9947
	1	1.0000	0.0000	0.0000	28	
5	0	0.9952	1.0000	0.9976	5233	0.9952
	1	1.0000	0.0000	0.0000	25	

**Table 7 sensors-25-02121-t007:** Summary statistics for model performance.

Parameter	Model Accuracy	Expected Error Rate
Path loss	0.9971	0.0029
Attenuation	0.9948	0.0052
Shadowing	0.9937	0.0063
Fading	0.9801	0.0199
Interference	0.9756	0.0244
Noise	0.9706	0.0294

**Table 8 sensors-25-02121-t008:** Comparative analysis of LoRa network optimization approaches.

Models	PDR	E (Jule)	Complex (ops)	Proc (s)
CNN-JCESD [5]	84.5%	0.50	5210^9^	0.1
CORA [10]	95%	50 J	100,000	43.4
MLR-CPLS [13]	91%	0.288607	111,057.0	0.690203
BSGP-BLSTM-SAE [6]	98.8%	0.552	1,638,400	1.638
SlidingChange [27]	47%	522.6	9.99	0.5438
ML-ADR [8]	99%	0.500	2,240,000	0.0310
Multi-task LSTM	100%	0.07987	17,216,000	0.00016

PDR, packet delivery ratio. E, energy consumption. Complex, computational complexity. Proc, processing time.

## Data Availability

The data supporting the reported results can be found at https://github.com/FatimahAlghamdi-KAU/Multi-task-Long-Short-Term-Memory.git (accessed on 17 February 2025).

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
