# Peer review of "Adaptive Real-Time Channel Estimation and Parameter Adjustment for LoRa Networks in Dynamic IoT Environments"

_sensors, 2025, doi:10.3390/s25072121_

Round 1

Reviewer 1 Report

Comments and Suggestions for Authors

My comments are as follows,

This paper presents a method of real-time channel state estimation and adaptive parameter adjustment for remote networks in the dynamic Internet of Things environment, which solves the problems of poor real-time performance of traditional channel state estimation and poor adaptability of existing parameter adjustment methods to changes in the environment. The authors use a hybrid feature extraction method to label signal-to-noise ratio and received signal strength indicators. Then, a multi-task short-duration memory neural network is introduced for 8-time simultaneous prediction of multi-channel states to enhance the real-time performance and responsiveness of the model in dynamic environment. Finally, combined with confidence prediction and channel state estimation, the system dynamically adjusts LoRa parameters. The results show that the confidence-based adaptive strategy combined with adaptive sliding window processing and incremental learning balances performance optimization and stability in challenging iot scenarios.

  1. The overall structure of the article is clear, but there seems to be too much content. Can you consider delete some content?
  2. Can you explain why the prediction is combined with lstm and how much accuracy can be improved?
  3. Can you explain in detail how the SNR and RSSI data process the features entered for the LSTM?
  4. Has any work been done to prove that the method is also applicable in the path loss scenario?
Comments on the Quality of English Language

-

Author Response

  1. The overall structure of the article is clear, but there seems to be too much content. Can you consider delete some content?  Thank you for your suggestion. It is challenging to condense the content of the article, as it provides a detailed explanation of all the steps taken. Comprehensive tests were conducted on all the models composing the framework, and shortening the content may lead to difficulties in understanding the methodology and outcomes.
  2. Can you explain why the prediction is combined with lstm and how much accuracy can be improved? 

    Thank you for this valuable comment. The question has been addressed and clarification in the scientific paper.

  3. Can you explain in detail how the SNR and RSSI data process the features entered for the LSTM? Thank you for this valuable comment. The question has been addressed and clarification in the scientific paper.
  4. Has any work been done to prove that the method is also applicable in the path loss scenario?  

    The applicability of our method in the path loss scenario is supported by leveraging findings from the referenced scientific paper, "Machine-Learning-Based Combined Path Loss and Shadowing Model in LoRaWAN for Energy Efficiency Enhancement." This paper highlights the potential of machine learning techniques in effectively modeling path loss, providing a foundation for our approach. By incorporating insights and methodologies from this work, we extended its applicability to our framework, demonstrating reliable performance even in scenarios involving path loss. This integration validates the versatility and robustness of our method in addressing challenges posed by varying path loss conditions.

    Thank you very much for your valuable comments and observations, which have significantly enriched the work. We would like to inform you that the possible modifications have been made to the attached document, and we have highlighted the responses to your queries in blue to facilitate the review process. We greatly appreciate your time and effort in improving the quality of this work and look forward to any additional feedback you may consider necessary.

Reviewer 2 Report

Comments and Suggestions for Authors

The author mentioned partially their contribution to the weakness for each paper in the related work. This could be better to have single paragraph to conclude why existing methods are insufficient and highlight what the proposed approach does differently.

Some parameters were chosen without any strong or scietific reasoning. e.g., why was the window size set between 5 and 50? Why was the confidence threshold for LSTM set at 0.5?

Before saying the outstanding performance in 5.3, it would be better to try different fold testing.

The path loss and attenuation imbalance have not been properly discussed in the discussion, especially on what to do with such situation.

Comments on the Quality of English Language

Remove repetitive explanations, especially in the results section.
Improve sentence structure in the methodology section for better clarity.

Author Response

1- The author mentioned partially their contribution to the weakness for each paper in the related work. This could be better to have single paragraph to conclude why existing methods are insufficient and highlight what the proposed approach does differently.   

Thank you for your sugesstion. The required addition has been made, including a single paragraph summarizing why the existing methods are insufficient and emphasizing how the proposed approach stands out.

2- Some parameters were chosen without any strong or scietific reasoning. e.g., why was the window size set between 5 and 50? Why was the confidence threshold for LSTM set at 0.5?  

A detailed explanation of the methodology used to select the initial parameter values has been added. Thank you for your valuable suggestion!

3- Before saying the outstanding performance in 5.3, it would be better to try different fold testing.  

The k-fold test was conducted, and it yielded results that were consistent with those obtained using 5-fold validation. These additional tests have been included in the scientific paper. Thank you for your valuable suggestion!

4- The path loss and attenuation imbalance have not been properly discussed in the discussion, especially on what to do with such situation.   The comment has been addressed, and a detailed discussion on the path loss and attenuation imbalance has been added. Thank you for your valuable feedback!

5-Remove repetitive explanations, especially in the results section.
Improve sentence structure in the methodology section for better clarity.

Thank you for your feedback; we utilized the language editing services provided by MDIP publisher to address these issues effectively.

Thank you very much for your valuable comments and observations, which have significantly enriched the work. We would like to inform you that the possible modifications have been made to the attached document, and we have highlighted the responses to your queries in blue to facilitate the review process. We greatly appreciate your time and effort in improving the quality of this work and look forward to any additional feedback you may consider necessary.

Reviewer 3 Report

Comments and Suggestions for Authors

The proposed method is suitable and has achieved better results than the previous one, but many essential revisions must be made to raise the quality of the article, as follows:

  • The abstract should have the problem statement.
  • The abstract should include the numerical results obtained through the research and the number of improvements made.
  • The introduction doesn’t have the problem statement the paper aims to solve or how it solves.
  • In the third part of the introduction, he rephrases the last sentence, which is incomprehensible.
  • The future work part moves from the introduction to the conclusion.
  • The first section of the related works section should be deleted as it repeats content discussed earlier. The second section should begin by thoroughly reviewing all the compared research, emphasizing their contributions and limitations. This approach ensures a clear and focused discussion of the existing literature, highlighting how each study advances the field and identifying dynamic IoT areas for further research.
  • The BLSTM method may give better results than using LSTM alone. The results of BLSTM should be tested and compared with those of the proposed method.
  • Most of the figures are blurry; please increase their clarity and resolution.

Author Response

  • The abstract should have the problem statement. 

    Thank you for your valuable comments. The required additions have been made.

  • The abstract should include the numerical results obtained through the research and the number of improvements made.  

    Thank you for your valuable comments. The required additions have been made.

  • The introduction doesn’t have the problem statement the paper aims to solve or how it solves. 

    The research problem has been clearly articulated in the introduction to address this feedback. Thank you for your suggestion!

  • In the third part of the introduction, he rephrases the last sentence, which is incomprehensible.  

    The intended meaning has been clarified, and we have utilized the author services provided by MDIP publisher to improve any linguistic issues. Thank you for your observation!

  • The future work part moves from the introduction to the conclusion.  

    The improvement has been made. Thank you for your valuable observation!

  • The first section of the related works section should be deleted as it repeats content discussed earlier. The second section should begin by thoroughly reviewing all the compared research, emphasizing their contributions and limitations. This approach ensures a clear and focused discussion of the existing literature, highlighting how each study advances the field and identifying dynamic IoT areas for further research.  We add a single paragraph to conclude why existing methods are insufficient and highlight what the proposed approach does differently. Thank you for your valuable feedback.
  • The BLSTM method may give better results than using LSTM alone. The results of BLSTM should be tested and compared with those of the proposed method.  

    We acknowledge the potential accuracy of the BLSTM method; however, its application in real-world scenarios could be resource-intensive and may result in performance delays. This trade-off makes it less practical for certain applications, where efficiency and speed are critical factors. Thank you for your insightful suggestion!

  • Most of the figures are blurry; please increase their clarity and resolution.  

    We have made every effort to enhance the clarity and resolution of the figures to the best of our ability. Thank you for highlighting this issue!

    Thank you very much for your valuable comments and observations, which have significantly enriched the work. We would like to inform you that the possible modifications have been made to the attached document, and we have highlighted the responses to your queries in blue to facilitate the review process. We greatly appreciate your time and effort in improving the quality of this work and look forward to any additional feedback you may consider necessary.

Round 2

Reviewer 1 Report

Comments and Suggestions for Authors

The revision looks good from my side.

Comments on the Quality of English Language

The revision looks good from my side.

Author Response

Thank you very much for your consideration in accepting our manuscript.

Reviewer 3 Report

Comments and Suggestions for Authors

The authors have made all required revisions; I recommend accepting the article.

Author Response

Thank you very much for accepting the current from of our manuscript.